# Improving subduction interface implementation in dynamic numerical models

Dan Sandiford[1,2] and Louis Moresi[1]

[1]School of Earth Sciences, University of Melbourne, VIC, 3010, Australia
[2]Institute of Marine and Antarctic Studies, University of Tasmania, TAS, 7004, Australia
**Correspondence:** Dan Sandiford (dan.sandiford@utas.edu.au)

**Abstract.** Numerical subduction models often implement an entrained weak layer (WL) to facilitate decoupling of the slab and upper plate. This approach is attractive in its simplicity, and can provide stable, asymmetric subduction systems that persist for many tens of Myr. In this study we undertake a methodological analysis of the WL approach, and use these insights to guide improvements to the implementation. The issue that primarily motivates the study is the emergence of significant spatial and temporal thickness variations within the WL. We show that these variations are mainly the response to volumetric flux gradients, caused by the change in boundary conditions as the WL material enters and exits the zone of decoupling. The time taken to reach a quasi-equilibrium thickness profile will depend on the total plate convergence, and is around 7 Myr for the models presented here. During the transient stage, width variations along the WL can exceed $4\times$, which may impact the effective strength of the interface, through physical effects if the rheology is linear, or simply if the interface becomes inadequately numerically resolved. The transient stage also induces strong sensitivity to model resolution. By prescribing a variable thickness WL at the outset of the model, and by controlling the limits of the layer thickness during the model evolution, we find improved stability and resolution convergence of the models.

## 1 Introduction

The process of stable asymmetric subduction requires that the down-going plate is substantially decoupled from the overriding plate along the subduction interface (Gerya et al., 2008). Determining the combination of processes that allow plate-bounding faults to develop is a longstanding problem in geodynamics (Lenardic and Kaula, 1994; Trompert and Hansen, 1998; Moresi and Solomatov, 1998; Tackley, 2000; Bercovici, 2003; Bercovici and Ricard, 2014). Even when the subduction interface is assumed *a priori*, the implementation within a continuum modelling framework is not trivial. Accordingly, a range of modelling approaches have been developed. Different approaches may reflect the style of model (e.g. instantaneous vs. long term), the degree to which multiphyics are represented (e.g. hydrogeologic processes), the subduction driving forces (kinematic vs. dynamic), as well as numerical method (finite difference vs. finite element methods vs. boundary element methods). A common approach in long term, dynamic models is to implement the subduction interface as a continuously-entrained layer of weak material (WL). Fig. 1a shows an example of this approach, implemented using Underworld2. This kind of approach

is relatively simple to implement, and can provide stable, asymmetric subduction regimes that persist for many tens of My, as shown in Fig. 1b. While the WL approach has been utilised in a broad range of studies, a detailed methodological analysis of this strategy is lacking (although valuable insights appear in Arcay (2017); Manea and Gurnis (2007); Arcay (2012); Čížková et al. (2002); Androvičová et al. (2013)). This study focuses on the behaviour of a constant viscosity WL implementation, within a 2d thermo-mechanical subduction setup. The results have relevance for the precision, efficiency, and reproducibility of dynamic subduction models.

## 2 The subduction interface

The subduction interface refers to the plate boundary fault at earth's convergent margins. Exhumed subduction interfaces are typified by melange zones, often 100's of meters in width, with coherent blocks embedded within a sedimentary and/or serpentinised matrix (Shreve and Cloos, 1986; Vannucchi et al., 2008; Kimura et al., 2012; Bebout and Penniston-Dorland, 2016). The subduction interface zone is characterised by rheological and petrological complexity, low strength, as well as the abundance of water and the critical role of fluid pressure (Abers, 2005; Audet et al., 2009; Bachmann et al., 2009; Gao and Wang, 2014; Hardebeck, 2015; Duarte et al., 2015; Bebout and Penniston-Dorland, 2016). Subducted sediments can reach depths of up to 80 km (Bayet et al., 2018), similar to the inferred depth at which the slab and mantle become coupled, and can be traced in the composition of arc magmas (Plank and Langmuir, 1993). The proportion of subducted sediment may lead to large variations in the mechanical properties in the deep subduction interface (Behr and Becker, 2018). Entrainment of upper plate material in a process known as subduction erosion also occurs at a substantial number of convergent margins (Huene and Scholl, 1991). Subduction interfaces therefore incorporate material from the subducting plate, the accretionary prism, and the upper plate (Vannucchi et al., 2008). However, the degree to which variability in these influxes impacts long-term subduction dynamics is debated (Cloos and Shreve, 1988; Schellart and Rawlinson, 2013; Duarte et al., 2015; Behr and Becker, 2018).

Fluid pressure plays a key role in subduction zones interfaces as it determines the effective frictional strength, controls the seismic/aseismic character of slip (Audet and Schwartz, 2013) and dictates the extent of hydrous mineral formation along the subduction interface (Reynard, 2013). In the shallow megathrust, pore fluid water and clay minerals are thought to exert a major control on the rheology (Vrolijk, 1990). Seismic imaging demonstrates the presence of fluids along the plate boundary (Audet et al., 2009). In some cases pore pressures are inferred to be near lithostatic values (Audet et al., 2009). While many subduction zone megathrusts are capable of hosting great earthquakes, there is a spectrum of behaviour from locked to creeping. Ongoing debate surrounds the strength implications of this divergent behavior (Gao and Wang, 2014; Hardebeck, 2015; Hardebeck and Loveless, 2018). The limit of the seismogenic zone can range from 5 to 50 kilometres in depth (Tichelaar and Ruff, 1993). The topography of most forearcs is consistent with average shear stresses of 15 MPa over the long-term (Lamb, 2006). Megathrust shear stresses in the range 1-100 MPa, with mean shear stresses also around 15 MPa, are consistent with results from heat flow studies (Peacock, 1996; Gao and Wang, 2014).

At greater depths, fluids are generated by devolatilization reactions (Bebout and Penniston-Dorland, 2016). The deeper part of subduction interface where the slab is in contact with the serpentinised upper plate mantle is thought to be controlled primarily by weak hydrous minerals which allow effective slab-mantle decoupling but also inhibits unstable seismic slip (Hirauchi and Katayama, 2013; Reynard, 2013). It remains unclear whether the rheology of the deep subduction interface is dominated by viscous or (stable) plastic behavior (Proctor and Hirth, 2016; Li and Ghosh, 2017). The thickness of subduction interfaces, particular in the deeper aseismic zone, is not well constrained. Estimates range from between tens of metres to a few kilometres (Abers, 2005; Cloos and Shreve, 1988; Vannucchi et al., 2008, 2012). Behr and Becker (2018) used structural relationships in exhumed subduction interfaces to infer the relative strength of different lithologies. Metasediments and serpentinites which commonly form the matrix within the subduction melange appear to be the weakest components.

## 3  Past modelling approaches

In efforts to study subduction zone dynamics, a range of modelling approaches have been developed. The subduction interface is a necessary model component whenever both subducting and upper plates are included. One approach is to incorporate the interface directly as a kinematic constraint into the simulation, i.e. by specifying continuous normal and discontinuous tangential velocities in the model solution surrounding the fault (Billen et al., 2003; Christensen, 1996). This idea was developed into a dynamic framework with finite element models that incorporate internal stress boundaries; the result is zero width faults within the continuum mechanical representation of the lithosphere (Zhong and Gurnis, 1992, 1995; Aagaard et al., 2008). Because moving mesh nodes are needed to capture proper fault advection, accurate tracking of large scale deformation is challenging. A more common approach is to apply finite regions of weak constitutive behavior within a static mesh. The velocity field naturally develops strain localization around the weak zone, although the 'faults' are usually very much broader than real plate boundaries. This approach was first applied as a spatially-fixed low viscosity zone that could decouple the plates but would not allow trench motion (e.g. Gurnis and Hager, 1988). This type of implementation was developed so that narrow low-strength weak zone 'stencils' could also be advected to allow trench motion (Kincaid and Sacks, 1997; Billen and Hirth, 2007).

Over the past decade the use of an entrained weak layer (WL) has become an increasingly common strategy (e.g. Babeyko and Sobolev, 2008; Capitanio et al., 2010; Magni et al., 2012; Chertova et al., 2012; Čížková and Bina, 2013; Garel et al., 2014; Holt et al., 2015; Agrusta et al., 2017; Glerum et al., 2017; Arredondo and Billen, 2017). Rather than using a fixed weak zone, the subduction interface is typically implemented by imposing a material layer at the top of the subducting plate that is advected with the flow and continuously entrained into the decoupling region. The WL approach enables a mobile trench while also helping to accommodate plate bending, which is important in the absence of a free surface. Additionally, it isolates the extremely weak parts of the system (the plate boundary) from the plates, enabling long-term asymmetric subduction to occur. In this study, we focus on a simple constant-viscosity implementation of the WL using the Underworld2 code, as described in Section 2.

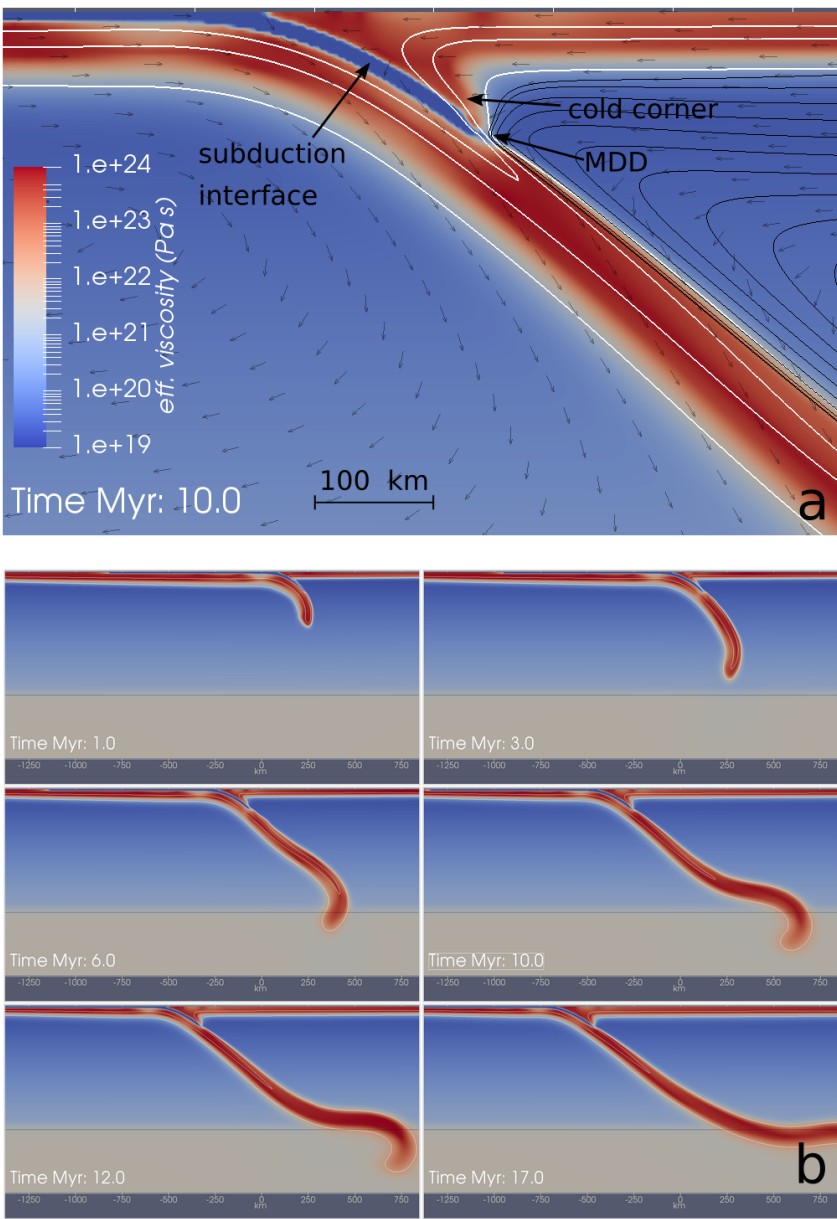

**Figure 1. Example of dynamic subduction simulation.** Colormap shows the effective viscosity on a logarithmic scale, and reveals the weak finite-width subduction interface which enables decoupling of the slab and the upper plate. The implementation of the interface follows a modified WL approach which we refer to as an embedded fault (EF). The subduction interface extends to 100 km depth, beyond which the slab and mantle become coupled (MDD). Above the MDD part of the mantle wedge tends to become stagnant due to the strong temperature dependence of the rheology. This is often referred to as a cold corner. a) closeup of slab in the upper mantle. White contour lines show 400, 700 and 1200°C isotherms. Black lines are streamlines of the velocity in a fixed upper plate reference frame. Arrows show model velocity field. b) Time evolution of the same model, showing slab rollback and interaction with the transition zone.

Although the WL is commonly implemented in numerical subduction models, there is some ambiguity as to whether this approach represents an attempt to explicitly model subduction interface dynamics, or should instead be conceived as an ad hoc solution to modelling the plate boundary. WL implementations do mimic several critical processes that are thought to contribute to the long-term weakness and stability of the subduction interface. One of these is the process of entrainment or 'self-lubrication' (Lenardic and Kaula, 1994). The entrainment of both sediments and fluids along the slab interface likely plays a key role in maintaining subduction interface weakness. Additionally, the WL approach might be linked with deformation-localizing processes such as damage, grain size reduction and fabric development. Indeed, WL models have been conceived as a limiting case in which comprehensive damage is assumed within the interface material and hence can be prescribed at the outset (Tagawa et al., 2007). Despite the fact that we can draw analogies to relevant physical processes, our view is that the WL approach constitutes an ad hoc solution to the sub-grid phenomena of plate boundary shear localisation.

The behavior of the WL is discussed is some detail in Arcay (2012, 2017). These studies highlight the tendency for the subduction interface to develop spontaneous thickness variation as the models evolve. Typically the interface widens near the trench, building a prism-like complex, and thins at depths beyond the brittle-ductile transition ( 50 kms). This pattern was also noted in the boundary element models of (Gerardi and Ribe, 2018), who attributed a downdip thickness variation to lubrication layer dynamics. The tendency for thinning will impact the level of mesh resolution required in a model (Arcay, 2017). Additionally, WL formulations are likely to evolve maximum subduction interface thickness around twice the imposed thickness, potentially impacting effective stress along the interface, as well as the accuracy of predicted thermal structure. Understanding and controlling these spontaneous width variations will be important in terms of using dynamic models to explore sensitive subduction-zone processes, such as metamorphism and melting near the slab top.

## 4  Methods

### 4.1  Numerical model setup

The numerical subduction models developed in this study represent the time evolution of simplified conservation equations for mass, momentum and energy within a 2D Cartesian domain. Fig. 2 provides an overview of the model domain, as well as initial and boundary conditions. The depth of the domain is 1000 km, and the aspect ratio is 5. Initial temperature conditions define two plates which meet at the centre of the domain, including a small asymmetric slab following a circular arc to a depth of 150 km. The subducting plate has an initial age of 50 Myr at the trench, while the upper plate age is 10 Myr. Both plates have a linear age profile with an initial age of zero at the sidewalls. This setup allows the model to evolve under the driving force of internal density anomalies (sometimes referred to as a fully dynamic model). Apart from the presence of a WL, there is no compositional difference between the subducting and upper plate, nor do we include any compositional differentiation within the oceanic lithosphere. The only aspects of the model setup that are varied are the details of subduction interface implementation (described in the following section) and the model resolution.

The mantle is treated as an incompressible, highly viscous fluid in which inertial forces and elastic stresses can be neglected. The mechanical behaviour of mantle (including the thermal lithosphere) is prescribed by a composite rheological model that includes a linear high-temperature creep law, as well as a scalar visco-plastic flow law, sufficient for capturing pseudo-brittle as well as distributed plastic deformation within the slab. Thermal buoyancy is the only source of density variation in the model.

The thermal variations are coupled to the momentum equation through their effect on density, which follows the Boussinesq approximation. A detailed description of the governing equations, constitutive laws, and physical parameters are given in Appendix A.

Approximate solutions to the incompressible momentum and energy conservation equations are derived using the finite element code Underworld2. Underworld2 is a Python API which provides functionality for the modelling of geodynamic processes.

Underworld2 solves the discrete Stokes system through the standard mixed Galerkin finite element formulation. The domain is partitioned into quadrilateral elements, with linear elements for velocity and constant elements for pressure ($Q_1/dP_0$)(Arnold and Logg, 2014). Material properties are advected on Lagrangian tracer particles. Unless otherwise stated, models have a mesh resolution of 160 elements in the vertical direction, refined to provide an element width of ∼3 km at the surface, and a particle density of 30 tracers per element. Particles are added and removed to maintain density near this value. During quadrature,

material properties are mapped to quadrature points using nearest-neighbour interpolation. The Lagrangian tracer particles are used to distinguish the subduction interface material from the rest of the system (lithosphere and mantle). Underworld2 solves the energy conservation equation using an explicit Streamline Upwind Petrov Galerkin (SUPG) method (Brooks and Hughes, 1982). In this approach, a Petrov-Galerkin formulation is obtained by using a modified weighting function which affects upwinding-type behaviour. The Stokes system has free-slip conditions on all boundaries. The energy equation has constant

(Dirichlet) and zero-flux (Neumann), on the top and bottom boundary respectively. The left and right sidewalls have a constant temperature equal to the mantle potential temperature (1673 K). The surface temperature is 273 K.

### 4.2   Subduction interface implementation

The focus of our study is a common approach to modelling the subduction interface, sometimes referred to as a weak layer or weak crust. This approach has two intrinsic features: 1) the material that provides decoupling within the interplate zone

is a distinct material type with rheology that contrasts with the plates/background material; and 2) the weak material layer is distributed along some or all of the subducting plate, so that the flow itself entrains new weak material into the decoupling region (Billen and Hirth, 2007; Garel et al., 2014; Čížková and Bina, 2013; Arredondo and Billen, 2017; Agrusta et al., 2017). In this study we use the abbreviation WL to refer to a typical implementation, namely one in which the distribution of weak material is fully self-evolving within the deforming subduction interface zone. We also demonstrate a modified version of this

approach, which we refer to as an embedded fault (EF). The EF implementation differs from the WL in that the width of the interface is constrained in terms of its minimum and maximum thickness. While this manipulation of the interface material could be achieved in different ways, our implementation is based on a reference line of particles, which lie at the base of the weak interface and are advected along with the material swarm. At each timestep we remap material types based on the relative

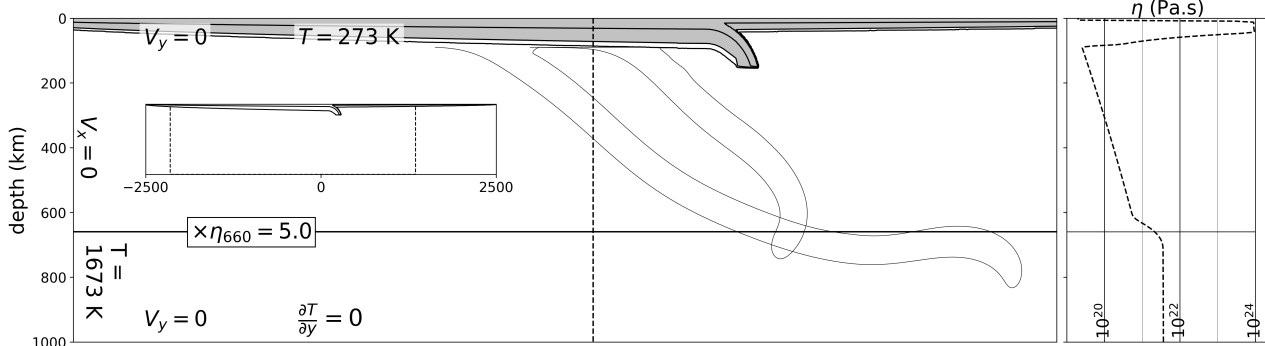

**Figure 2. Initial conditions for 2D thermo-mechanical subduction models**. All models in this study have a 50 Myr initial slab age. Velocity boundary conditions are free-slip on all walls of the domain (zero tangential stress). The dark gray region in the main panel shows parts of the model colder than 1100 °C. Inset shows the full domain. Outlines show the slab position at 10 and 20 Myr. Right hand panel shows viscosity profile evaluated along the vertical dashed line in main panel.

position of the material particles with regard to the reference line. The relationship between the EF reference line, the material swarm, and the mesh is shown in Fig. 3. In the EF approach, we also initialise the interface with a non-uniform thickness. For reasons discussed in the following sections, the thickness of the interface material within the decoupling zone should be close to double the thickness that is prescribed on the top of the subducting plate.

5 It is important to emphasise that differences between the WL and EF implementations relate only to the distribution of weak material within the model. All other aspects of the interface representation remain identical. In each case the interface has a constant viscosity. Weak material is continually prescribed along the upper-most part of the subducting plate as it moves away from the ridge, with a specified constant thickness ($W_{\text{init}}$ = 10 km). The upper plate does not contain any weak/interface material. For simplicity, there is no rheological variation between the shallow (frictional) megathrust and the deeper, viscous

10 interface. To control the upper limit of the MDD, a depth-dependent cosine taper is used to transition the subduction interface rheology to the background mantle rheology. The taper for the transition (both WL and EF) begins at 100 km, and has a width of 30 km. Decoupling is strongly inhibited at depths greater than the taper onset. Hence the depth of the taper onset (100 km) effectively controls the upper limit of MDD. Likewise, at a distance of 800 km from the trench, along the subducting plate, the rheology of the interface material transitions to background mantle rheology. This avoids interaction of weak material with the

15 spreading ridge.

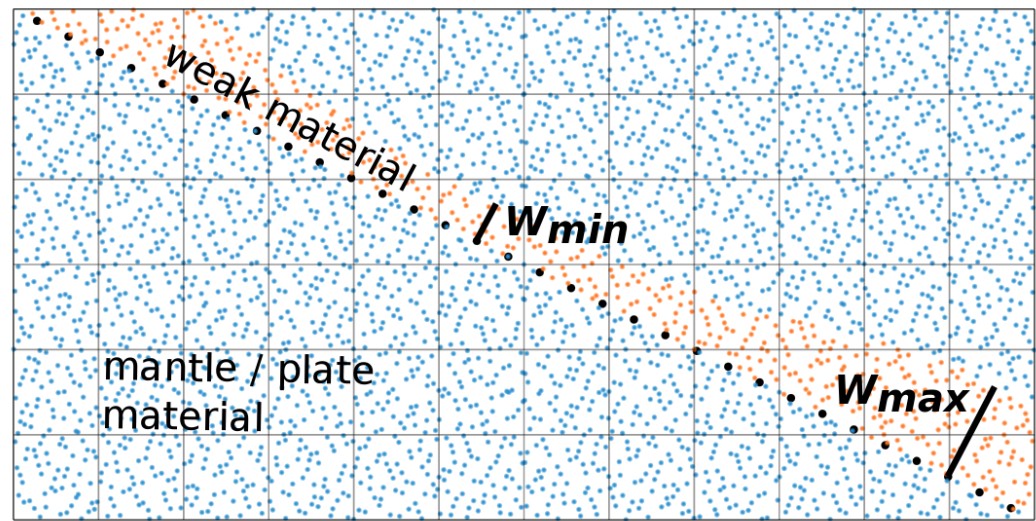

**Figure 3. Schematic of the embedded fault (EF) implementation**. The EF is a modification to the standard weak layer approach (WL). As with the WL, shear localizes due to a finite-width layer of weak material (represented by orange particles). A reference line of tracer particles (black points) is advected with the flow, at the base of the weak material layer. This line provides a reference to enforce width limits on the weak material, denoted by $W_{min}$ and $W_{max}$. At each timestep we remap material types based on the relative position of the material particles to the reference line. Note that the element size here is not shown to scale.

## 5   Analysis of modelling approaches

### 5.1   The weak layer approach

Fig. 4 shows the evolution of the subduction interface thickness over a 20 Myr period, based on the WL setup described in section 4. This model reveals the spontaneous development of substantial WL thickness variations, not only near the trench (i.e. near the accretionary prism) but throughout the entirety of the decoupling zone. Previous studies have commented on the occurrence of similar thickness variations (Arcay, 2017; Gerardi and Ribe, 2018) although their origin has not been closely considered. We propose that such variations arise primarily from the way that the kinematics of the flow in the WL effect the downdip volumetric flux. In the following discussion we refer to a slab-based coordinate system in a fixed upper plate reference frame, where $\hat{y}$ is the direction orthogonal to the slab midplane, $\hat{s}$ is the direction parallel to the slab midplane, and the plate convergence velocity is equal to the subducting plate velocity ($V_s$). Before reaching the trench, the WL material travels with the subducting plate, with near-uniform velocity and negligible shear ($\frac{\partial V_s}{\partial \hat{y}} \approx 0$). In the subduction interface zone, however, the WL

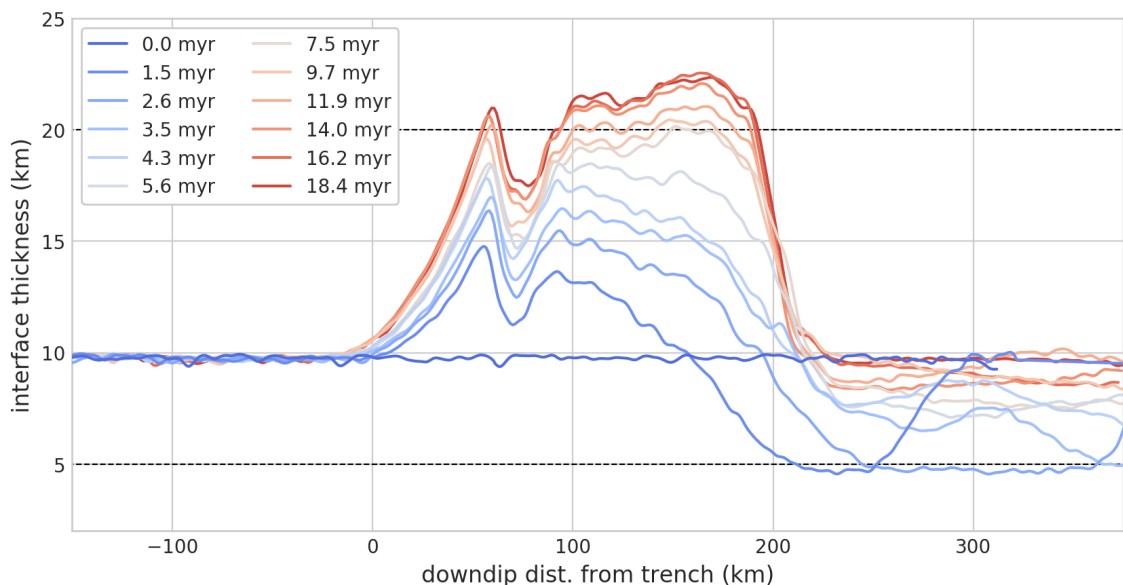

**Figure 4. Interface thickness evolution in WL implementation**. The initial layer thickness ($W_{\text{init}}$) is 10 km. Colored lines show the evolving thickness plotted as function of downdip distance from the trench. Dashed black lines show the expected maximum (equilibrium) and minimum (transient) thickness, based on a kinematic description of the flow evolving from a uniform thickness weak layer.

material decouples the slab and upper plate, and velocity gradients across the interface are finite. For a Newtonian rheology, the gradient is expected to be linear with $\frac{\partial V_s}{\partial \hat{y}} \approx \frac{V_s}{W}$ (i.e. Couette flow). This change in velocity profile across the WL means that there is a smaller volumetric flux (per unit length normal to the interface) compared to the flux of material being delivered on the incoming subducting plate. To illustrate this point we consider a simple boundary-driven Stokes flow, as shown in Fig.
5, which provides a useful analogy to the flow in the WL. In this model, flow is driven by a constant tangential velocity on the lower boundary, while the upper boundary of the model has a patch of frictional (no slip) nodes, surrounded on both sides by a free-slip boundary condition. The step change in boundary conditions imposes volumetric flux gradients causing the weak layer (shown in blue) to thicken near the start of the no slip region, and thin near the end. The width changes of the weak layer proceed until the overall volume flux reaches equilibrium.

Returning to our subduction model results, the validity of our description can be tested by considering the thickness profile
10 required for volume flux equilibrium. Assuming that the decoupling zone maintains a Couette profile, independent of the local thickness, flux equilibrium requires that the thickness of the WL in decoupling zone will be twice that imposed on the incoming plate. This prediction, based on a simplified kinematic description of flow in the interface is broadly consistent with the long-term evolution observed in the WL models (e.g. Fig. 4).

While our discussion so far has mainly emphasised the transition of the WL from the oceanic part of the subducting plate into
15 the decoupling zone, similar processes take place where the slab and the mantle become coupled (the MDD). When a constant

interface width is prescribed, the interface thickness near the MDD initially decreases. Again, this is because the volumetric flux increases as Couette flow in the decoupling region transitions to the fully coupled flow. The location where the subduction interface thins, provides a proxy for the MDD. A complexity arise due to the fact that the MDD in our model is not constant. Instead the MDD tends to become deeper as the corner of the mantle wedge cools and stagnates. This means that the effective boundary conditions on the flow within in the interface are also time varying. This accounts for why the location of interface thinning migrates downdip as time progresses, as shown in Fig. 4.

Based on analysis of typical WL setup, we argue that the primary thickness variations reflect a simple response to changes in volumetric flux along the interface. In addition to these flux-controlled changes, Fig. 4 shows that the subduction interface also develops a persistent short-wavelength width perturbation just beneath the tip of the forearc at a downdip distance of $\sim$ 60 - 80 km. Here the interface thins by $\sim$ 3 - 4 km. This occurs in combination with strong, localised plastic deformation in the upper plate. It seems likely that such short-wavelength anomalies may be affected by a range of factors, including the interface and lithosphere rheology, mesh resolution, and material advection scheme. If so, these features are likely to be somewhat model dependent, in contrast to the to long-wavelength, flux-related thickness variations which reflect intrinsic kinematics of flow in the WL setup.

## 5.2 Improving the weak layer approach

The embedded fault (EF) implementation, described in Section 4, consists of two complimentary strategies. Firstly, we may initiate the WL with a variable thickness. Secondly we can control the WL thickness throughout the simulation, by remapping WL material (particles) to background material, and vice versa. Given these controls, perhaps an obvious first issue to address is what happens if we simply enforce a constant width interface at all times. While this would in some ways be a desirable approach, doing so results in the development of a very spurious subduction morphology. Fig. 6b shows results from such a case ($W_{\mathrm{max}} = W_{\mathrm{min}} = W_{\mathrm{init}} = 10$ km). After 15 million years of model evolution a very atypical subduction morphology has developed, with an extremely low angle megathrust beneath a forearc region with a width of greater than 600 km. While geologically irrelevant, the example shown in Fig. 6b provides a useful insight into the way in which the flow in the interface can influence model dynamics. When the subduction interface is forced to remain at constant width, the interface is unable to evolve towards flux equilibrium. Persistent interface-normal velocity components result, and the compounding effect eventually distorts the morphology of the entire subduction hinge region.

Fig. 6a shows the slab morphology developed by instead using $W_{\mathrm{max}} = 1.9 \times W_{\mathrm{init}} = 19$ km (we discuss the choice of 1.9 in the next section). The subduction morphology in this simulation is more realistic, and consistent with outcomes from a standard WL approach. The evolution of the interface thickness from the same simulation is shown in Fig. 5.2. In addition to controlling the width of the interface throughout the EF simulation, we have also prescribed the initial interface thickness in the decoupling zone (from the trench to a depth of 100 km depth) to have value equal to $W_{\mathrm{max}}$ (19 km). In this way, we have tried to preemptively impose a thickness profile closer to the flux equilibrium. Fig. 5.2 shows that this strategy reduces, but

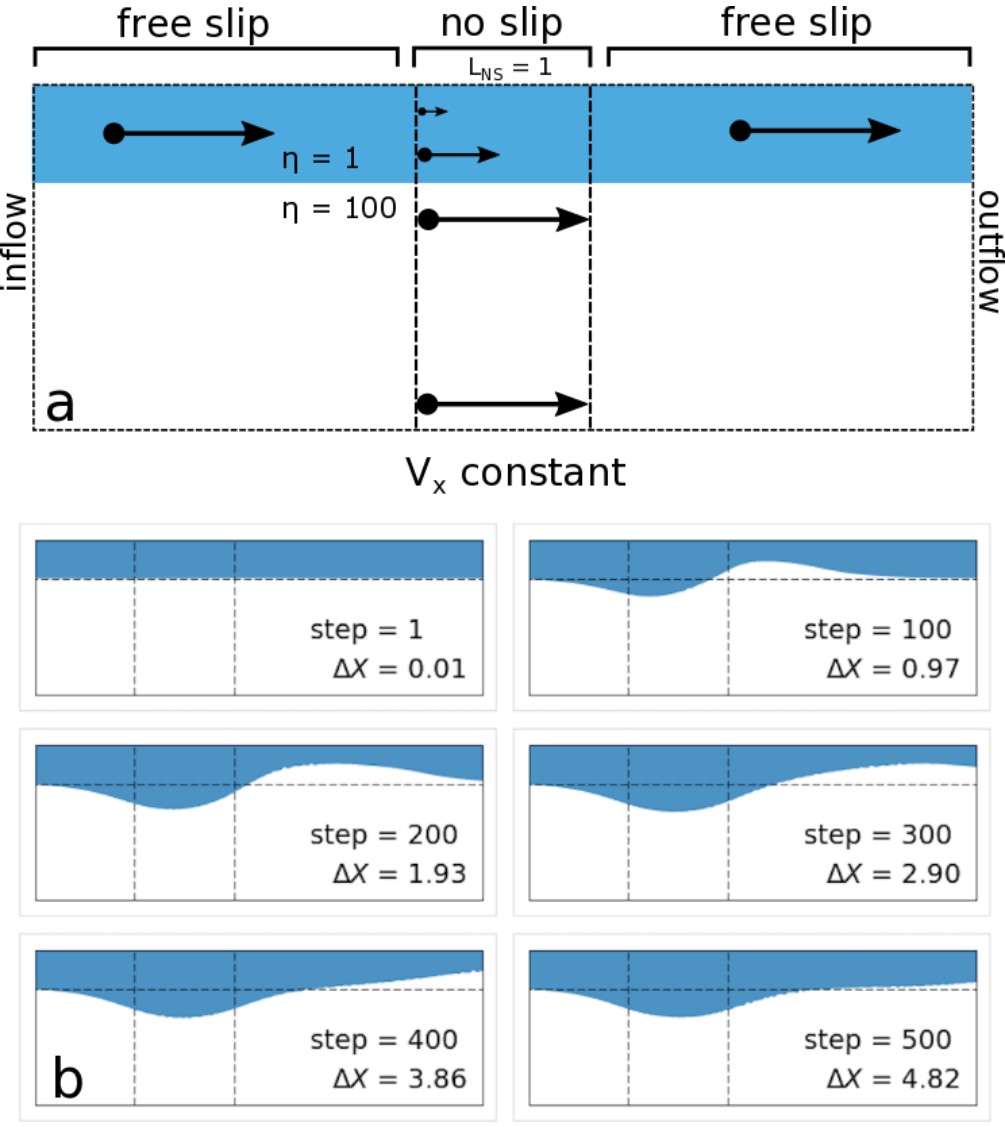

**Figure 5. Analogue for WL thickness evolution**. a) model setup for a boundary-driven flow where a weak layer (shown in blue) interacts with a stronger layer (white). The top surface of the model is free slip, except for a patch of no-slip nodes ($V_x = 0$). The bottom surface has a constant horizontal velocity component. b) evolution of material in the boundary-driven model. $\Delta X$ represents the accumulated displacement along the bottom boundary, normalised by the width of the no-slip patch.

does not fully eliminate, the transient stage of interface adjustment. It is difficult to fully eliminate the transient stage because the equilibrium interface thickness profile is party determined by the MDD, which is not constant. The evolution of the MDD is a response to the cooling of the mantle wedge and the development of a stagnant 'cold corner' (see Fig. 9). Prescribing initial temperature conditions that include the cold corner, would be one way to further reduce the amount the transient adjustment of the interface. Fig. 7 shows the state of the material swarm in the EF model, compared to the equivalent WL model. This reveals a typical distribution of interface material once a quasi-equilibrium thickness has been established.

In addition to the thickness variations related to volumetric flux, the WL approach can develop short wavelength thickness variations, as seen in Fig. 4. In the EF implementation, we found that using a value of $W_{\mathrm{max}}$ slightly less than $2.0 \times W_{\mathrm{init}}$ helps to suppress short wavelength thickness variations, without significantly effecting the overall model evolution. In other words, while we need to allow the interface to develop some amount of thickness variation, it may be advantageous to use a value slightly less than $2 \times W_{\mathrm{init}}$. Fig. 10 shows results from a number of experiments where the value of $W_{\mathrm{max}}$ was changed. Qualitatively, we see that that the models have very similar long term evolution once $W_{\mathrm{max}}$ is greater that $1.75 \times W_{\mathrm{init}}$.

## 5.3 Stability and convergence

So far we have discussed results based on well-resolved models, with 160 elements across the 1000 km vertical domain. At this resolution, the vertically-refined mesh provides 3.2 elements within the subduction interface (at $W_{init} = 10$ km). Note that this increases to more than 6 elements in the decoupling zone, once the interface has thickened to $\sim 20$ km. We now look at the convergence of models under variable resolution, based on a standard WL approach as well as the EF implementation. Fig. 11 shows the slab temperature field at 10 Myr for a set of models with varying resolution: 72, 96, 128, 160, 192 elements in the vertical dimension. The ratio of the initial interface width ($W_{\mathrm{init}}$) to the element width has values of 1.4, 1.9, 2.6, 3.2, 3.8 for the respective mesh resolutions. The dashed slab outline in Fig. 11 shows the morphology for a simulation with 192 elements in the vertical dimension; this is used as a reference model in the following analysis. Note that while we have varied the mesh resolution, all models have the same spatial particle density. Fig. 11a shows results using a standard WL implementation. At the lowest resolution (72 elements) the simulation stalls and the slab undergoes runaway thermal decay. At 96 elements, the model is still strongly impacted by under-resolution of the subduction interface. Fig. 11b shows equivalent results using the EF implementation. Qualitatively, we see that EF models are more stable at lower resolution. For instance, the EF model at lowest resolution (72 elements) more closely reproduces the evolution of the reference model than does the WL model with 96 elements.

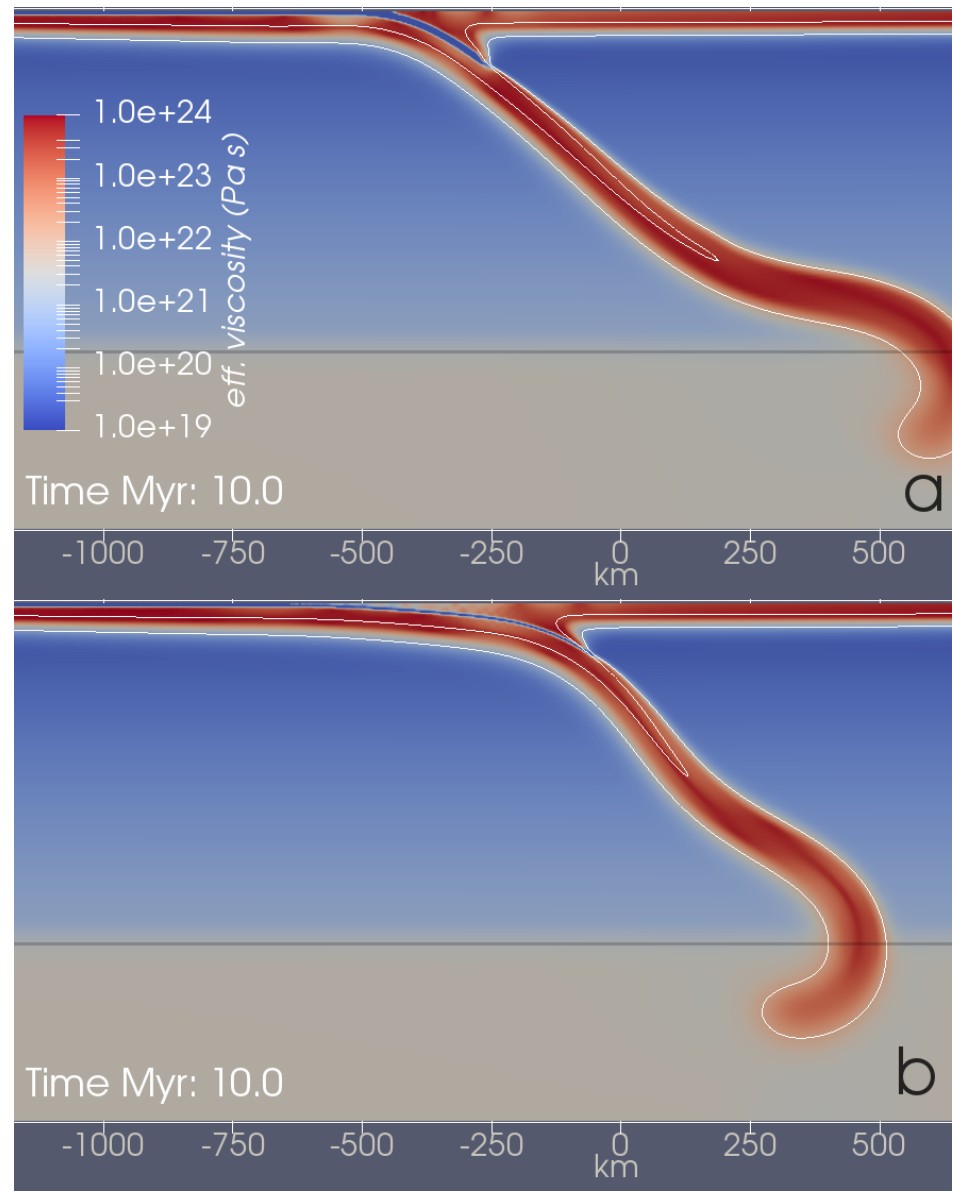

**Figure 6. Effect of variable maximum $W_{\mathrm{max}}$ in EF models**. The colormap shows the effective viscosity at a model time of 10 Myr. White lines show isotherms at 700 and 1200 °C. a) shows EF model with $W_{\mathrm{max}} = 1.9 \times W_{\mathrm{init}}$ = 19 km. b) shows EF model with $W_{\mathrm{max}} = W_{\mathrm{init}} = 10$ km. In this case, due to the fact that the subduction interface cannot adjust its thickness, an extremely long, low-angle interface develops, with forearc distances of $\sim$ 500 km.

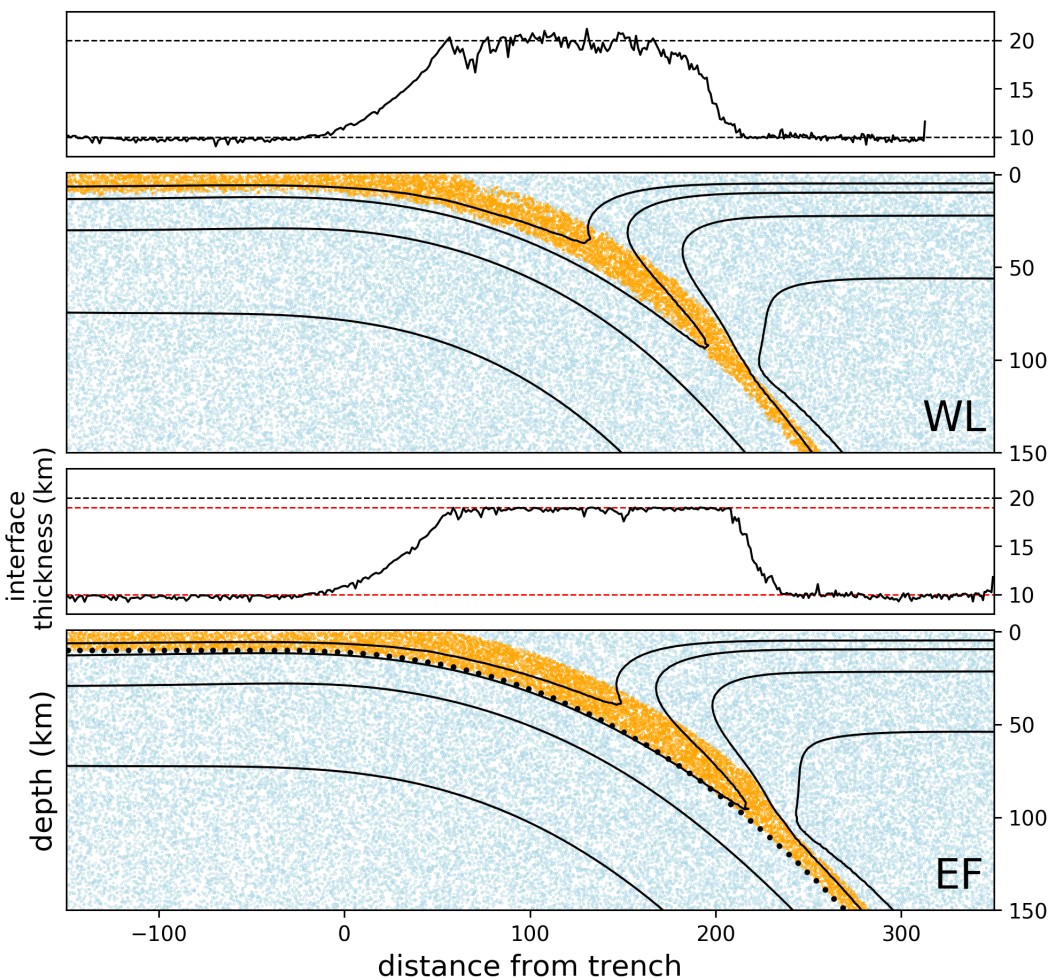

**Figure 7. Distribution of subduction interface material**. Results from a standard WL model are shown in the top panel, EF model shown in bottom panel. Both models have a constant viscosity interface rheology. Model time is 12.5 Myr in both cases. In the lower panels, orange points are the subduction interface material, blue points are the background material (mantle/lithosphere). Points in the material swarm and along the EF reference line have been down-sampled for clarity. Solid black lines over the material points are isotherms. Smaller panels show the measured interface thicknesses in each case; red horizontal lines show the thickness constraints $W_{\min}$, $W_{\max}$.

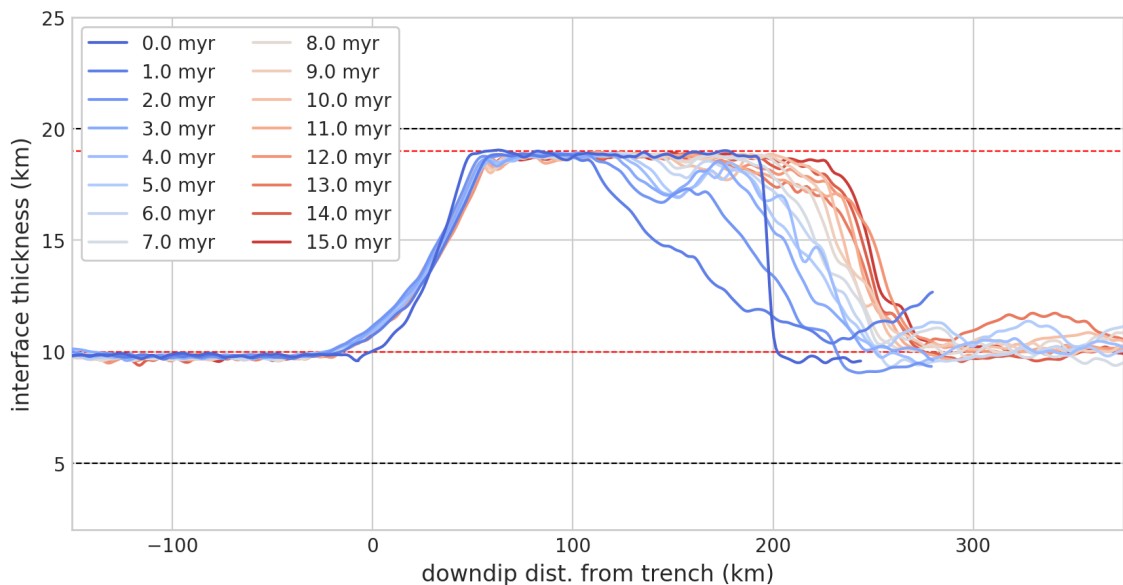

**Figure 8. Interface thickness evolution, EF approach**. In the EF implementation limits are imposed in the maximum and minimum thickness, as shown by the red dashed lines. Unlike the standard WL approach, the initial thickness of the interface is variable. The initial thickness of the interface within the decoupling region is equal to $W_{\mathrm{max}}$. The black lines show typical range in which a WL model will vary, assuming a constant initial thickness of 10 km (see also Fig. 4). The results shown here refer to the same model as shown in Fig. 1 and Fig. 6.

Fig. 11 suggests that the EF models converge more closely with increasing resolution. We quantify this by tracking the relative error ($L_2$) of the temperature field in the lower resolution models with respect to the reference model (192 elements). The relative error with respect to the reference model ($T_{\mathrm{ref}}$) is:

$$E = \left[ \frac{\int_{\Omega} (T - T_{\mathrm{ref}}) \cdot (T - T_{\mathrm{ref}}) dV}{\int_{\Omega} T_{\mathrm{ref}} \cdot T_{\mathrm{ref}} dV} \right]^{1/2} \tag{1}$$

Fig. 13 shows relative error results for the same set of models shown in Fig. 11, confirming more rapid resolution convergence in the EF models relative to the standard WL approach. For most of the models shown in Fig. 13 the relative error accumulates rapidly in the first 7-10 million years of the simulation, while the error rate flattens after this. This is similar to the time taken for the WL interface to reach its equilibrium thickness (e.g. Fig. 4). This suggests that models are strongly resolution-sensitive during the transient phase of the interface adjustment. At low resolution (72, 96, 128 elements), errors in both WL or EF models express this sensitivity. Interestingly, at 160 elements, the EF case exhibits nearly-constant error accumulation during the model evolution. This suggests that we have succeeded in reducing the sensitivity of the subduction interface implementation, relative

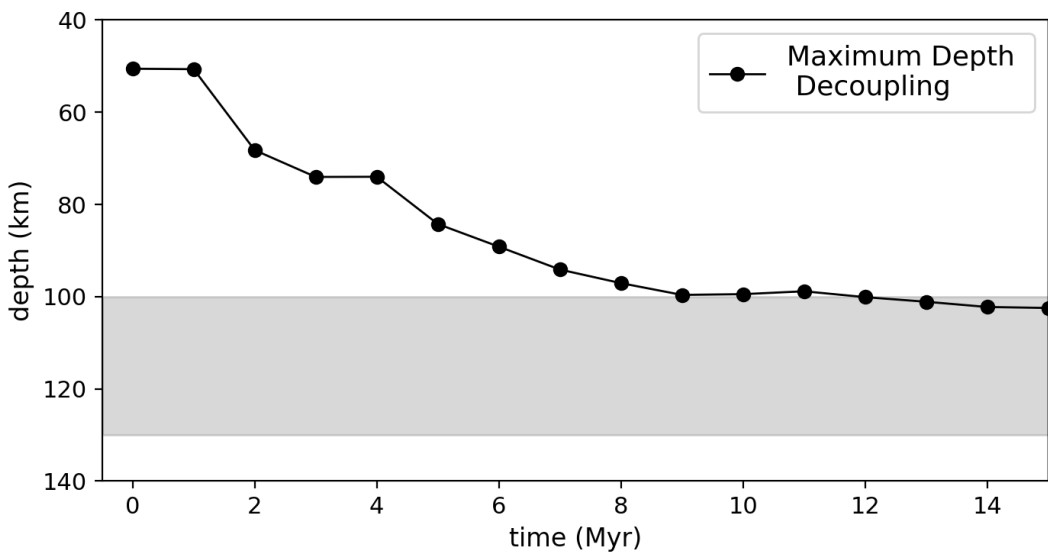

**Figure 9. Evolution of the maximum depth of decoupling (MDD).** As the mantle wedge cools, it progressively stagnates, driving deeper decoupling within the subduction interface. The grey region in the figure shows the depth interval over which the subduction interface transitions to the background mantle rheology as prescribed with a cosine taper (see Section 4). Results are from the same model as shown in Fig. 6a, i.e. an EF model with $W_{\mathrm{max}} = 1.9 \times W_{\mathrm{init}}$

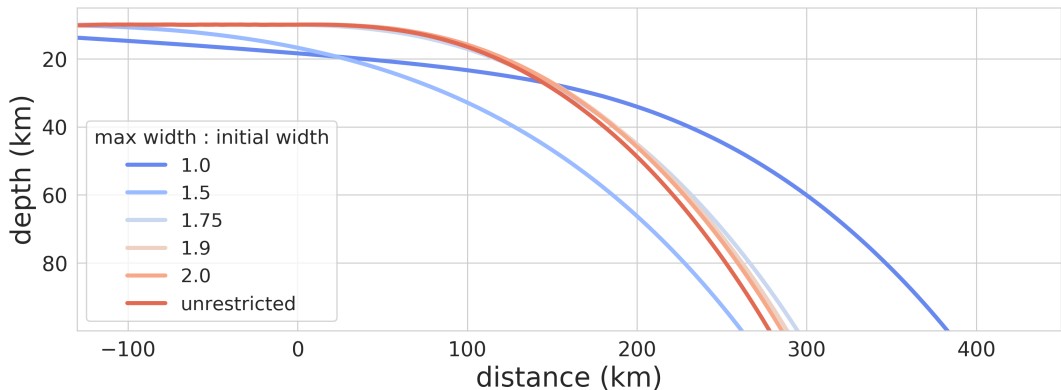

**Figure 10. Embedded fault models (EF) with variable $W_{\mathrm{max}}$.** Lines show the morphology of the base of the subduction interface (the EF reference line) at 10 Myr for different models with varying $W_{\mathrm{max}}$. When the interface thickness variation is strongly constrained, the evolution of the model is significantly effected. See also Fig. 6.

to the overall error accumulation rate. The latter may be influenced by additional factors which we have not controlled for here, such as the timestep size in the advection-diffusion implementation which is controlled by the mesh resolution).

Fig. 12 shows the distribution of interface material in the lowest-resolution WL model (shown in the top left panel in Fig. 11a). In this case, under-resolution of the weak layer induces strong coupling between the slab and the upper plate at relatively shallow depths and begins to thin the WL, due to the development of strong flux gradients (as described in Section 5). This induces further coupling and yet more thinning. This proves to be catastrophic feedback process, causing the simulation to stall and enter runaway thermal decay. The EF approach provides stability in this context, by inhibiting the feedback cycle. While this behavior is mainly relevant for models run at low resolution, the increased stability of the EF approach is a useful property, particularly from a model development perspective.

## 6 Discussion and Conclusions

The entrained weak layer (WL) is a common approach for implementing the subduction interface in long-term dynamic simulations. We have discussed aspects of WL implementation that can have an unintended impact on model evolution. The first involves the transient evolution of a uniform thickness interface to a variable thickness - uniform flux, configuration. If not properly accounted for, thinning of the deeper part of the WL could lead to numerical under-resolution, as previously suggested (Arcay, 2017). If the WL has a viscous rheology, the thinning will lead to higher stresses. This can induce a positive feedback when higher stresses increase the amount of partial coupling, inducing further thinning. Even for seemingly well-resolved models, the transient behavior of the subduction interface appears to be responsible for strong mesh sensitivity and poor resolution convergence. In general, models with plastic/frictional rheologies should be less sensitive to these transient adjustments, as the stress should not depend on the width of the interface.

Another tendency of the WL models is to develop persistent short wavelength thickness variations. These may represent interface instabilities, as are observed in Couette flows past a deformable boundary (Shankar and Kumar, 2004). These tend to dominate on the shallow part of the boundary with the upper plate. While flux-related thickness variation will be expected for any model implementation of WL, boundary instabilities (short wavelength) are likely to be more variable across different codes. They may depend on additional details of implementation, such as the rheology of the plates, material advection and interpolation schemes. Together, these issues are likely to hinder efforts to produce reproducible results between codes.

These unintended behaviors of the WL approach can be partly mitigated by controlling the thickness of the interface. We demonstrate a simple implementation of this concept, which utilises a line of reference points at the base of the WL, allowing us to remap material types based on proximity to this line. We call this approach an embedded fault (EF). The ability to constrain the thickness of the interface improves the resolution convergence of numerical models, as well as the stability at lower resolutions. The EF does add complexity to models, both in the sense of implementation as well as the introduction of new parameters to control the specific details (e.g. $W_{\mathrm{max}}$). There are obviously other implementation strategies that could be

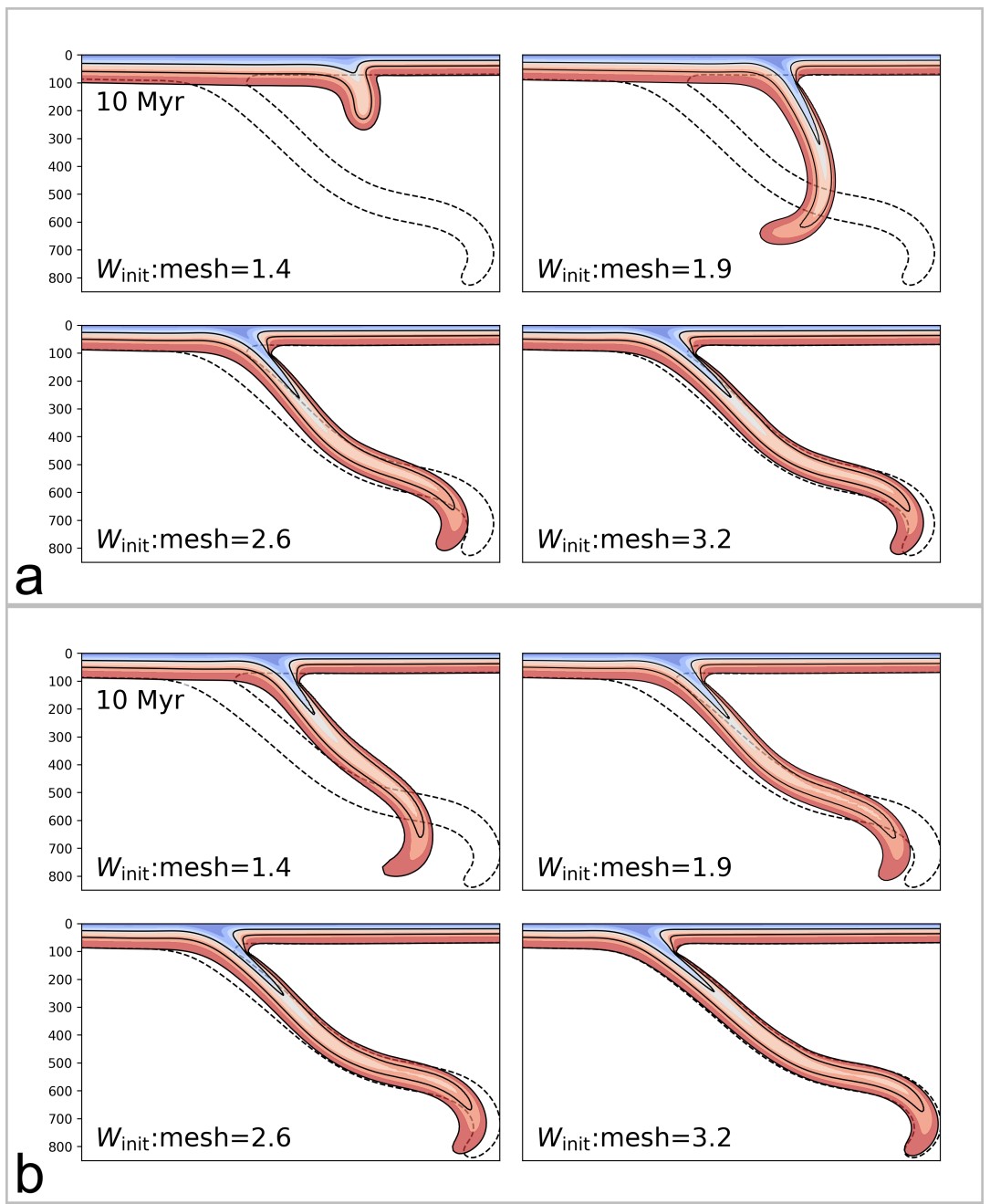

**Figure 11. Implementation comparison with variable resolution.** Colormap shows slab temperature field at 10 Myr, masked above 1250 °C. Each figure shows a series of models run at different resolution. The total number of elements in the vertical coordinate was 96, 127, 160, 192. Corresponding subduction interface resolution is displayed in the figure, representing the initial fault thickness ($W_{init}$) divided by the local element size. a) models using standard WL implementation. b) models using the EF implementation.

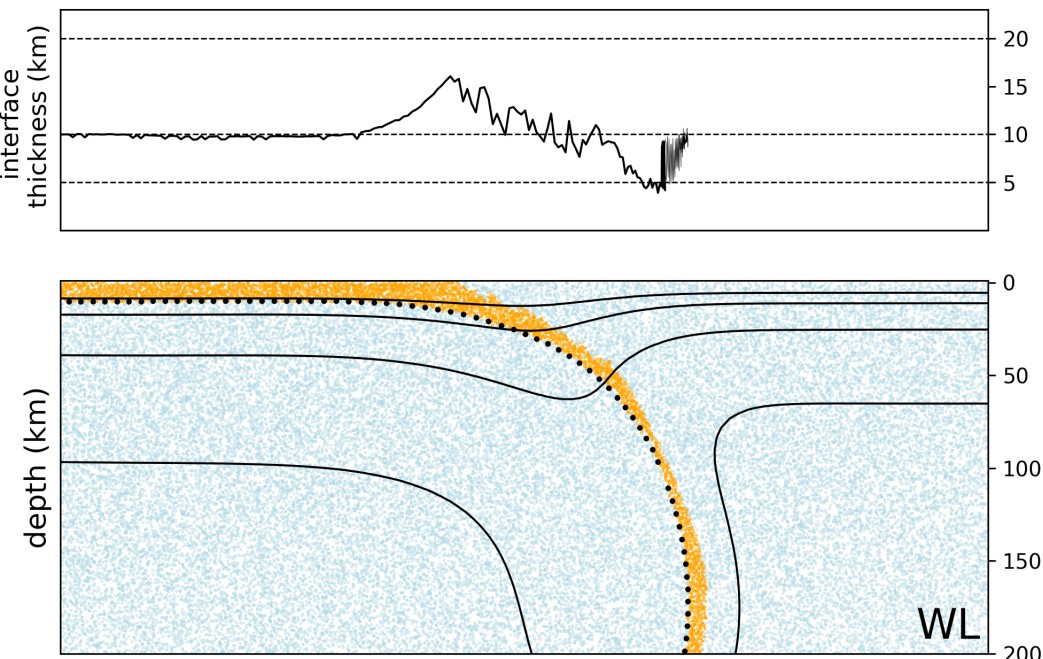

**Figure 12. Interface thickness in under-resolved WL model.** In the lower panel orange points show the subduction interface material, blue points are the background material (mantle/lithosphere). Upper panel shows interface thickness. Figure shows the model state 10 Myr after model initiation. This model is also shown in the upper-left panel of Fig. 11a.

developed in order to achieve similar outcomes. These should be explored in future studies. Overall, this study provides a better understanding of the behaviour of subduction models utilising WL approaches. These insights offer a basis for achieving better outcomes in terms of model reproducibility and precision.

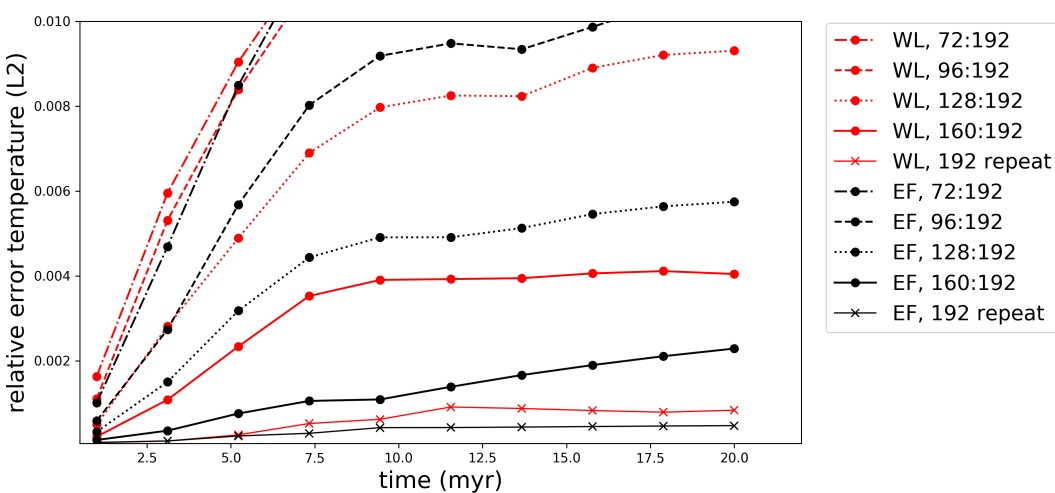

**Figure 13. Convergence of models with varying resolution.** Vertical axis shows the relative error ($L_2$) of the temperature field. We have truncated the vertical axis so as to focus on trends in the higher-resolution results. The error value is based on the error between the highest resolution model (192 elements in the vertical axis) and each of the models with lower resolution, as labelled in the figure. Experiments were repeated at the highest resolution to provide a baseline for model reproducibility (labelled 'repeat').

## Appendix A: Governing equations, constitutive relationships and physical parameters

### A1 Continuity, momentum and energy equations

On geological time scales the Earth's mantle behaves as a highly viscous, incompressible fluid, in which inertial forces can be neglected. The flow caused by internal buoyancy anomalies is described by the static force-balance (momentum conservation) and continuity equations:

$$\sigma_{ij,j} + \rho g_i = 0 \tag{A1}$$

$$u_{i,i} = 0 \tag{A2}$$

where $u_i$ is the $i^{th}$ component of the velocity. Repeated indices denote summation, and $u_{,i}$ represents partial derivative with respect to the spatial coordinate $x_i$. The full stress tensor appearing in Eq.A1 can be decomposed into deviatoric and mean components:

$$\sigma_{ij} = \tau_{ij} + p\delta_{ij} \tag{A3}$$

It is noted that sign of the pressure ($p$) is opposite to the mean stress tensor, consistent with the convention that fluid flows from high to low pressure. The deviatoric stress tensor ($\tau$) and the strain rate tensor ($D_{ij}$) are related according to the constitutive relationship:

$$\tau_{ij} = 2\eta D_{ij} = \eta(u_{i,j} + u_{j,i}) \tag{A4}$$

Substituting Eqs.A4 & A3 into Eq.A1 gives the Stokes equation, which involves two unknown variables: pressure, and velocity. The Stokes and continuity equation are sufficient to solve for the two unknowns, together with appropriate boundary conditions. An approximate solution to these equations is derived using a Galerkin Finite Element method, implemented in the *Underworld2* code.

The thermal evolution of the system expresses the balance between heat transport by fluid motion, thermal diffusion and internal heat generation by the 1st Law of Thermodynamics, assuming incompressibility:

$$\rho C_p \frac{DT}{Dt} = q_{i,i} + \rho Q \tag{A5}$$

where T is the temperature and Q is the heat production rate (everywhere zero in this study). Diffusion rates are described by Fourier's Law, which satisfies the 2nd Law for positive conductivity ($k$):

$$q_i = -kT_{,i} \tag{A6}$$

Inserting Eq.A6 into Eq.A5, and using the definition of the material derivative gives:

$$\frac{\partial T}{\partial t} + u_i T_{,i} = (\kappa T_{,i})_{,i} + \frac{Q}{C_p} \tag{A7}$$

where $\kappa = \frac{k}{\rho C_p}$ is the thermal diffusivity.

The thermal variations are coupled to the momentum equation through their effect on density. At pressures in planetary interiors, silicate minerals are weakly compressible and this is generally considered as a perturbation to an incompressible flow. The Boussinesq approximation accounts for the buoyancy forces while neglecting the associated volume change allowing us to assume incompressibility (Eq. A2). In the case of density variations due to temperature, the equation of state is:

$$\rho = \rho_0(1 - \alpha(T - T_p)) \tag{A8}$$

where $\rho_0$ is the density at a reference temperature (here the mantle potential temperature $T_p$). $\alpha$ is the coefficient of thermal expansion. It is generally much smaller than one, making the Boussinesq approximation reasonable.

The equations and parameters that appear in the numerical models are based on equivalent dimensionless forms of the governing equations. We use the following characteristic scales (e.g Christensen, 1984):

$$\bar{x}_i = x_i \left[\frac{1}{d}\right], \ \bar{u}_i = u_i \left[\frac{d}{\kappa}\right], \ \bar{\eta} = \eta \left[\frac{1}{\eta_0}\right], \ \bar{\tau} = \tau \left[\frac{d^2}{\kappa \eta_0}\right], \ \bar{t} = t \left[\frac{\kappa}{d^2}\right], \ \bar{T} = T \left[\frac{1}{\Delta T}\right], \tag{A9}$$

where $d$ is the mantle depth, $t$ is time, $\eta_0$ is the reference viscosity and $\Delta T = (T_s - T_p)$, is the superadiabatic temperature difference across the fluid layer. Substituting dimensional terms for scaled dimensionless values (e.g $x \rightarrow \bar{x}d$ ), and rearranging allows us to write the Stokes equation as:

$$2\bar{\eta}\bar{D}_{ij,j} + \bar{p}_{,i} = Ra(1 - \bar{T})(-\delta_{iz}) \tag{A10}$$

Overbars in Eq.A10 represent dimensionless quantities, and all dimensional parameters are contained in the dimensionless ratio $Ra$, the Rayleigh number which can be interpreted as a ratio of advection and diffusion timescales:

$$Ra = \frac{\rho_0 g \alpha \Delta T D^3}{\eta_0 \kappa} \tag{A11}$$

The dimensionless viscosity, which has a functional dependence on the total pressure, the temperature and the second invariant of the stress tensor, are described below.

The dimensionless form of the heat conservation equation is:

$$\frac{\partial \bar{T}}{\partial \bar{t}} + \bar{u}_i \bar{T}_{,i} = (\bar{T}_{,i})_{,i} + \bar{Q} \tag{A12}$$

where the dimensionless internal heating is given by:

$$\bar{Q} = Q \left[ \frac{d^2}{\kappa C_p \Delta T} \right] \tag{A13}$$

## A2 Rheology

Mantle silicates deform through a range of mechanisms. The most important high-temperature creep mechanisms are diffusion creep (low stress), which results in a linear relationship between stress and strain that is strongly dependent on grain size; and dislocation creep (high stress), which leads to a power law relationship between stress and strain that is independent of grain size. In addition to high-temperature creep, some form of stress limiting behaviour is expected to occur at low temperature, and high-stress. Glide-controlled dislocation creep (or Peierls creep), which includes a stress dependence of the activation energy, is likely to play a role, particularly in the cold part of subducted slabs Karato (2012). Nevertheless, it remains unclear whether Peierls creep allows sufficient weakening, as geophysical constraints on slabs would imply (e.g. Jain et al., 2017; Krien and Fleitout, 2008; Alisic et al., 2010). In geodynamic calculations, the effect of the Peierls mechanism is similar to that of plastic, temperature-independent plasticity models (Agrusta et al., 2017; Garel et al., 2014; Čížková and Bina, 2013).

To keep the models as simple as possible we include two deformation mechanisms: grainsize-independent diffusion creep and a plastic yielding based on a truncated Drucker-Prager plasticity model. The plastic strain rates are capable of representing brittle failure at low pressure (near the surface) and low-temperature plasticity at high pressure (deep within the slab). The value of the yield stress limit is chosen on the basis of previous studies, and is consistent with several lines of evidence suggesting that slabs to not support more than a few hundred MPa (Richards et al., 2001; Tackley, 2000; Watts and Zhong, 2000; Krien and Fleitout, 2008; Alisic et al., 2010). The viscosity associated with each deformation mechanism is combined using a harmonic average,

denoted by $\eta_c$. The entire computational domain, except for the subduction interface, is governed by the same composite rheology: there is no compositional distinction between the mantle and plates.

Ductile flow laws for silicates often have an Arrhenius temperature and pressure dependence, controlled by the activation energy $E$, and activation volume $V$ (Hirth and Kohlstedt, 2004). Additional dependencies, such as grain size and melt fraction, are neglected in this study, resulting in the following diffusion viscosity:

$$\eta_d = A \exp\left(\frac{E + p_l V}{R T_a}\right) \tag{A14}$$

where $p_l$ indicates the lithostatic component of the pressure, and $A$ is a constant. A linearised adiabatic term is added to the dimensionless temperature field, whenever it appears in an Arrhenius law:

$$T_a = T + z \times T_{,z}$$
$$T_{,z} = \frac{-\alpha g T_p}{C_p}$$

The dimensionless form of the creep law applied in the models uses the following scalings:

$$\bar{E} = E\left[\frac{1}{R\Delta T}\right], \quad \bar{W} = V\left[\frac{\rho_0 g d}{R\Delta T}\right], \quad \bar{A} = A\left[\frac{1}{\eta_0}\right] \tag{A15}$$

Note that $V \rightarrow \bar{W}$ includes a change from pressure dependence (dimensional) to depth dependence (dimensionless): The dimensionless diffusion creep viscosity can be written:

$$\bar{\eta}_d = \bar{A} \exp\left(\frac{\bar{E} + \bar{z}\bar{W}}{\bar{T}_s + \bar{T}_a}\right) \tag{A16}$$

where $\bar{z}$ is the dimensionless depth and $\bar{T}_s$ is the dimensionless surface temperature. The dimensionless linearised adiabatic component is incorporated as follows:

$$\bar{T}_a = \bar{T} + \bar{z} \times \bar{T}_{,\bar{z}}$$
$$\bar{T}'_{,\bar{z}} = T_{,z}\left[\frac{d}{\Delta T}\right]$$

The parameters chosen for the diffusion creep law are consistent with those derived from experimental data on dry olivine (Karato and Wu, 1993), providing an average upper mantle viscosity close to $1 \times 10^{20}$ Pa s. The relatively high value of the

activation volume produces relatively low viscosity asthenosphere $\sim 0.3 \times 10^{20}$, relative to the transition zone $\sim 5 \times 10^{20}$. Additionally, a viscosity increase ($\times \eta_{660}$) is applied at the 660 km discontinuity, consistent with inferences based on the geoid (Hager and O'Connell, 1981). For ($\times \eta_{660} = 10$), the lower mantle just beneath the 660 km is 50 times more viscous than the mean viscosity of the upper mantle. The parameters chosen produce radial viscosity profiles that are slightly higher than the

'Haskell constraint' ( $\eta_{\mathrm{mean}} = 1 \times 10^{21}$) over the upper 1400 km of the mantle (e.g. Becker, 2017).

A range of pseudo-brittle and plastic deformation mechanisms can be approximated in the fluid constitutive model by allowing non-linearity in the viscosity ($\eta = \eta(T, p, J_I, ...)$). The rheological model itself should be defined independently of the coordinate system, so it is necessary to define the constitutive model in terms of stress invariants ($J_I$). The standard viscoplastic approach (Spiegelman et al., 2016) defines an effective plastic viscosity $\eta_p$ such that the deviatoric stress tensor is bounded by

a yield stress $\tau_y$:

$$\tau_y = 2\eta_p D_{ij} \tag{A17}$$

Assuming that $\eta_p$ is isotropic and scalar (i.e. eigenvectors of the strain-rate tensor and deviatoric stress are identical), one can use the magnitude of both sides to define the scalar effective plastic viscosity as:

$$\eta_p = \frac{\tau_{y(\mathrm{II})}}{2\epsilon_{\mathrm{II}}} \tag{A18}$$

where the subscript II, denotes the square root of the tensor second invariant.

The yield stress function in the computational models is a truncated Drucker-Prager criterion:

$$\tau_y = \min(\tau_{\mathrm{max}}, \mu p + C) \tag{A19}$$

where $\mu$ is the friction coefficient, and $C$ is the cohesion. The Drucker-Prager yield surface is defined by the full pressure $p$. Because the pressure that appears in the dimensionless Stokes equation (Eq. A10) is a dynamic pressure ($\bar{p}$), due to density

variations only, the lithostatic pressure (a function of vertical coordinate) needs to be accounted for. The dimensionless form of the yield stress is given by

$$\bar{\tau}_y = \min(\bar{\tau}_{\mathrm{max}}, \bar{\mu}(\bar{p} + \bar{p}_l \bar{z}) + \bar{C}) \tag{A20}$$

where:

$$\bar{\mu} = \mu,$$
$$\bar{C} = C\left[\frac{d^2}{\kappa\eta_0}\right],$$
$$\bar{\tau}_{\max} = \tau_{\max}\left[\frac{d^2}{\kappa\eta_0}\right], \tag{A21}$$
$$\bar{P}_l = \bar{z}\left[\frac{\rho_0 g d^3}{\kappa\eta_0}\right],$$

The effective plastic viscosity (dimensionless) is given by:

$$\bar{\eta}_p = \frac{\bar{\tau}_{y(II)}}{2\dot{\epsilon}_{II}} \tag{A22}$$

5   The final (composite) viscosity is the harmonic average of the viscosity associated with creep and plastic yielding:

$$\bar{\eta}_c = \frac{\bar{\eta}_d \bar{\eta}_p}{\bar{\eta}_d + \bar{\eta}_p} \tag{A23}$$

## A3   Model parameters and scaling values

This section provides a record of model parameters and reference values that are used in the models. The dimensional parameters quoted here are non-dimensionalised using the scaling system described in Appendix A and reference values provided in
10   Table A1. This scaling system is identical for all models used within the thesis.

| Reference value | Value | Symbol | Units |
|---|---|---|---|
| length | 2900 | $d$ | km |
| viscosity | $1 \times 10^{20}$ | $\eta_0$ | $\mathrm{Pa\,s}$ |
| density | 3300 | $\rho_0$ | $\mathrm{kg\,m^{-3}}$ |
| thermal diffusivity | $1 \times 10^{-6}$ | $\kappa$ | $\mathrm{m^2\,s^{-1}}$ |
| gravity | 9.8 | $g$ | $\mathrm{m\,s^{-2}}$ |
| temperature | 1400 | $\Delta T$ | K |
| gas constant | 8.314 | $R$ | $\mathrm{J\,mol^{-1}\,K^{-1}}$ |
| Rayleigh Number | $3.31 \times 10^8$ | Ra | - |

**Table A1. Reference values** used to non-dimensionalise the Stokes and Energy equations, as described in Appendix A

| Parameter name | Value | Symbol | Units |
|---|---|---|---|
| domain depth | 1000 | - | km |
| domain width | 5000 | - | km |
| potential temp | 1673 | $T_p$ | K |
| surface temp | 273 | $T_s$ | K |
| viscosity min. | $1 \times 10^{18}$ | - | Pa s |
| viscosity max. | $1 \times 10^{24}$ | - | Pa s |
| diffusion creep volume UM** | $5.27 \times 10^{-6}$ | $V$ | $m^3 \, mol^{-1}$ |
| diffusion creep energy UM | 316 | $E$ | $kJ \, mol^{-1}$ |
| diffusion creep constant UM | $1.87 \times 10^9$ | $A$ | $Pa^n \, s^1$ |
| diffusion creep volume LM*** | $1.58 \times 10^{-6}$ | $V$ | $m^3 \, mol^{-1}$ |
| diffusion creep energy LM | 210 | $E$ | $kJ \, mol^{-1}$ |
| diffusion creep constant LM | $1.77 \times 10^{14}$ | $A$ | $Pa^n \, s^1$ |
| DP* friction coefficient | 0.1 | $\mu$ | - |
| DP cohesion | 20 | $C$ | MPa |
| yield stress max. | 200 | $\tau_{max}$ | MPa |
| sub. interface thickness | 10 | $W_{init}$ | km |
| sub. interface max. thickness | 19 | $W_{max}$ | km |
| sub. interface min. thickness | 10 | $W_{min}$ | km |
| sub. interface viscosity | $5 \times 10^{19}$ | - | Pa s |
| sub. interface depth taper start | 100 | - | km |
| sub. interface depth taper width | 30 | - | km |
| slab age at trench | 50 | - | Myr |
| slab radius of curv. | 200 | - | km |
| initial slab depth | 150 | - | km |
| upper plate age at trench | 10 | - | Myr |
| lower mantle viscosity increase | 15 | - | - |
| adiabatic temp. gradient | $3.7 \times 10^{-4}$ | - | - |
| internal heating | 0.0 | $Q$ | $W.m^{-3}$ |

**Table A2. Dimensional model parameters**: * Drucker-Prager, ** Upper Mantle, *** Lower mantle. Typical model element resolution was $800 \times 160$.

*Competing interests.* TEXT

The authors declare that no competing interests are present.

*Acknowledgements.* This research was partially funded by the Australian Government through the Australian Research Council Discovery grant DP150102887. Development of the Underworld2 code used in the simulations supported by AuScope. DS's postgraduate research at the University of Melbourne was supported by a Baragwanath Geology Research Scholarship. The study benefited from the authors' attendance of the CIDER Summer programs in 2016 & 2017 (funded by NSF grant EAR-1135452). This work was supported by resources provided by The Pawsey Supercomputing Centre with funding from the Australian Government and the Government of Western Australia. This research was supported by use of the Nectar Research Cloud, a collaborative Australian research platform supported by the National Collaborative Research Infrastructure Strategy (NCRIS).

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
