# Peer review of "Improving subduction interface implementation in dynamic numerical models"

_Solid Earth, 2019_

## Referee Comment (RC1) · Thibault Duretz (Referee) · 25 Feb 2019

This study aims at providing a new approach to model long-term subduction processes, particularly at the plate interface. The authors investigate the widely-used 'weak layer' (WL) approach and identifies some of its limitations. In this light, they propose an alternative approach, termed 'embedded fault' (EF), and show that this approach remedies some limitations of the WL approach. This work is interesting because many geodynamic modelers studying subduction processes are facing these issues. The outcome of this study (i.e. the EF approach) might help subduction modelers designing their models. It will surely help informing the community about the caveats of subduction modeling. That said, I was personally not convinced of using the proposed EF approach. The essence of this approach is to remap the geometry of the plate interface

at each time step of a simulation. The idea is thus to overrule the geometries predicted by the numerical simulation in order to facilitate interplate decoupling. I would personally not encourage code users to interfere within simulations by using ad hoc rules. I would rather expect an alternative solution that would not require interfering with an ongoing numerical simulation. This would necessitate an augmented modeling framework, by accounting for thermo-mechanical feedbacks (e.g. Thielmann and Kaus, 2012) hydrological (e.g. Dymkova and Gerya, 2013) or chemical processes) or an advanced rheological model (e.g. Bellas et al, 2018). This is likely beyond the scope of the current study, however this is a fundamental issue. ad these aspects are so far not really discussed. Prior to publication, I would hence recommend the authors to enrich their discussion - potentially around the above-mentioned points. I found the discussion very short and mostly restricted to the difference between WL/EF approaches. I would also encourage the authors to make sure that the figures are cited in increasing order. You'll find below a list of comments and suggestions.

Best regards, Thibault Duretz

p.1 l. 8 - What is "fully dynamic"? What are the requirements for a model to be "fully dynamic"? p.2 l.5 - "geodyanamics"

p.3 l. 15 - What is "full thermal modelling"? How can a thermal model be "full"?

p.5 l.9 - "an solution" p.5 l.11 - brackets around citation p.5 l.25 - no brackets around citation p.5 l.26 - a more important issue is when the interface locally thins out and becomes unresolved p.5 l.31 - & ?

figure 1 caption: The layer representing the subduction interface appears to have viscosity variations (top-left) while it is described as having a constant linear viscosity. Is figure 1 relevant at all, I don't see it called in the text?

p.6 l.15 - psuedo-brittle p.6 l.24- what function you use to refine the mesh? p.6 l.24- what is the horizontal resolution? p.6 l.30 - on all side p.6 l.33-34 - not clear wether left

and right hand side wall are treated similarly.

figure 2 caption: Since you mention normal velocity to be zero at boundaries, you may also add that the tangential shear stress is zero.

p.7 l.3 - "Past studies..." I would add references or delete this statement. p.7 l.8 - "Throughout this study, ..." this sentence reads weird or incomplete.

p.8 l.1 - "In this chapter" - is this part of a thesis or report? p.8 l.4 - Fig. 8 is called right after Fig. 3

From Section 4.3 and Figure 3 caption, it is not entirely clear what the principle of EL is. Do you mean that: (1) you pre-define a channel geometry that will remains constant during model evolution (2) you remap, at each timestep, material types based on their relative position of the particles with regard to the channel? If yes, please state it clearly.

p.8 l.7 "In a number of previous studies..." Do you mean variations in space or time? No sure wether the referencing is sound.

p. 10 l. 20 "effected" p. 10 l. 21 "these feature" p. 10 from l. 25 on. Better write $W_{\mathrm{min}}$ instead of $W_{min}$. Same for 'max' and 'init' p. 10 l.30 please add a scale on this figure p. 10 l.30 "physically inconsistent" do you mean geologically irrelevant? Is this so irrelevant by the way? p 10. l.34 - how would a free surface affect this behaviour?

figure 6 caption: what are red and black dashed lines? Which model is depicted here? The model presented in Fig. 7a or 7b?

p.12 l.10 - "reduce the amount the transient adjustment" p.12 . l.16 - "One advantage of the EF approach is that it offers improved precision in determining the thickness of the subduction interface." This is confusing, I thought the EL approach was aiming at imposing this thickness. How can it help to determine a thickness when it already imposes it?

p.12 l.17 - "Such precision will be important for studying highly pressure- and temperature-sensitive processes, such as metamorphism and melting near the slab top." Well, sure. This new parametrization will pre-define everything related to it. figure 7: very difficult to appreciate the dimensions, a scale is missing. What is the grey line? I think the notion of "physically-consistent subduction morphology" does not make sense.

figure 8: How is this MDD monitored? Do you measure the stress differences across the plate interface?

figure 9: do you use the same scale in x and y? figure 9 caption: "effected"

figure 10: at what time do these snapshots correspond? Do you use any particle reseeding?

figure 10 - 11: Given the fact that the mesh is not distorted, adding element edges would help the reader to realise how well the interface is resolved.

p.15 l.9 - you mentioned you ran simulations with "72, 96, 128, 160 elements". You can also include 192 elements as I understand.

p. 16 l.5 - "better" sounds very qualitative. You mean that low resolution models using a WL fails at capturing plate decoupling.

figure 13: I don't understand, why do repeated models at the resolution of 192 produce any error? If the models were repeated you should obtain the exact same results, don't you?

p. 20 l.10 - a mistake here, a mean stress is not lithostatic. It can be split into dynamic and lithostatic components.

p. 22 - Why using a lithostatic pressure field in the viscosity expression. Are obtained numerical solution of the actual pressure field not accurate enough to be used in the rheology?

References

Marcel Thielmann, Boris J.P. Kaus, Shear heating induced lithospheric-scale localization: Does it result in subduction?, Earth and Planetary Science Letters, Volumes 359–360, 2012, Pages 1-13,

Dymkova, D., and T. Gerya (2013), Porous fluid flow enables oceanic subduction initiation on Earth, Geophys. Res. Lett., 40, 5671‐5676doi.

Ashley Bellas, Shijie Zhong, David Bercovici, Elvira Mulyukova, Dynamic weakening with grain-damage and implications for slab detachment, Physics of the Earth and Planetary Interiors, Volume 285, 2018, Pages 76-90,

---

## Referee Comment (RC2) · Anonymous Referee #2 · 13 Mar 2019

This paper explores different parameterizations of a weak layer that is used to model one-sided subduction. It particularly explores the controls on the width of this weak layer and finds that, within the context of the models presented, there is a preferential width. In general I think this is a nice, albeit technical, contribution to the literature describing fully dynamical modeling of subduction zones. I have a few somewhat broader comments and a fair bit of more detailed comments.

Section 2 contains a fair bit of information that was already mentioned in the introduction or other superfluous. A bit of careful editing can shorten this section and make it clearer.

The citations in the text don't seem to follow any chronological or alphabetical order (see example in P4.l9-l11).

[Figure]

The weak fault seems rather wide (10 km as mentioned on page 7). If you have a 2 km grid resolution then that it is probably a good idea to smear it out over such a distance but what are your thoughts about FEM grid refinement around this fault to be able to bring it to more realistic thicknesses?

I wonder if the improvements you find when choosing the initial and maximum width of the weak zone to be 20 km is just a matter of the numerical resolution (mesh and #particles) that you use. Maybe you need at least 10 elements in this zone for things to stabilize. You mention that a lot of small wavelength variations disappear compared to using 10 km. This might be a symptom of discretization issues. It would be very helpful if you could repeat your exercise with higher resolution. I suspect your maximum width will go down with increasing resolution. In 5.3 you do a divergence test but I am not sure that this is very meaningful given that you show that coarser resolution meshes lead to worse results. It is far more exciting to see what happens when you start converging.

Figure 11. I am not certain it is useful to show underresolved results.

More detailed remarks (I wish SE would use continuous line numbering instead of restarting it every page. . ..).

P1 footnote

1. why footnote?

2. might be good to add a few original references as to why the slab-mantle wedge coupling appears to start at 80 km (that seems to be the case in most SZs; please just clarify why we think this is the case)

P2.l5. typo

P2.l25. typo

P2.l25-27. Full sentence here. You can probably rephrase this a bit better as in that sediments may be important amidst various other issues controlling plate velocities.

P2.l33. 10's -> tens

P3.l12-14. I am confused about this full sentence here. The seismogenic zone is generally not characterized by ending in the mantle (maybe between slab crust and mantle, but not slab mantle and mantle).

P3.l14. Average stresses: where?

P3. line starting at l22. This seems to take a single point of an antigorite-out boundary as some extreme. It's a bit more variable/complex than that. You can probably delete this sentence.

P4. Figure 1. last line "representative" of what?

P4.l14 missing article

P4.l5 missing plural

P4.l9 typo

P5.l11 typos

P5.l25. What is 'over-resolving'Âă?

P5.l30 just benchmarking

P5.l31. "1 km, &20K" Not sure what this means. The benchmark study referenced here showed that finite element models agreed on temperature predictions along the slab surface to within about 1K for 1 km resolution. There were some finite difference models that had larger differences. And here and below. It is just '20K' not '20\degreeK'.

P7 three sentences starting at l20. Don't you say essentially the same thing here? You might be able to condense this into a single sentence.

P8.l1 I hope a cosine taper is easy to implement in many subduction models. Is it not?

P9.l12ff. Maybe use subscript 'conv'?

 P9.bottom. MDD reference is repetitive

P10.l6 You are citing Figure 8 way out of order

P10.l21 typo

---

## Referee Comment (RC3) · Hana Cizkova (Referee) · 2 Apr 2019

Manuscript presents a detailed analysis of the properties of entrained weak layer that is often used in numerical models of subduction to decouple the plates. The authors demonstrate that this weak layer is changing its width both spatially and with time. After entering the subduction channel at the trench it increases width and then thins again as the subducting plate is coupled with the mantle flow below the overriding plate. This thinning, that may temporarily result in a decoupling weak layer actually yet thinner than the prescribed initial thickness, may result in locking of the subduction in case of lower resolution and certainly affects the subduction evolution. The authors explain this phenomenon using an analogy with a boundary-driven Stokes flow in a two layer material with boundary condition changing from free-slip to no-slip and back.

[Figure]

They then introduce an improvement to the standard weak layer approach (called here embedded fault) where the thickness of the weak layer is controlled and modified during the subduction evolution. This approach prevents transient thinning of the lower part of the decoupling layer and potential coupling of the subducting and overriding plates in case of lower resolution.

I find this paper very interesting. The weak layer approach is often used in subduction modelling while the numerical aspects of the implementation of this decoupling are seldom discussed or even mentioned. I therefore very much appreciate this systematic evaluation of the problem and suggested solution using ad hoc control of the layer thickness. The crutial point of the paper is made in fig. 12 where the authors illustrate that in their embedded fault approach low resolution has much smaller effect than in standard weak layer approach. This paper thus provides the reader with a recipe how to tackle the problem of decoupling the plates with an entrained weak layer – one may either use the suggested embedded fault approach, or use resolution high enough to resolve the thinned interface in the transient stage at the beginning of subduction. Alternatively, as just briefly mentioned in the discussion, this problem may be suppressed by nonlinear rheology, but that of course brings other complexities into the play.

The manuscript is nicely and clearly written and the topic fits the scope of Solid Earth, therefore I recommend it for publication. I only have couple of suggestions for mostly minor corrections.

1. You may perhaps explicitly mention that the weak layer has constant viscosity. The reader has to dig up this information by combining the sentence stating that except of subduction interface the rest of the domain is deforming according to composite rheology (at the end of paragraph 20 page 22 in Appendix) and the information in Table A2 (unless I overlooked this information somewhere else).

2. I don't see the logic in the order of figures – they are sometimes ordered and referenced rather randomly. Figure 1 is not referenced in the text.

3. Page 2, par. 25, line 4: remove of

4. Page 5, par. 5, line 5: an solution -> a solution

5. Page 5, par. 10, line 3: though -> thought

6. Page 6, par. 5, line 3: a initial -> an initial

7. Page 6, par. 20, line 5: a element -> an element

8. Caption Figure 11: Fig. a -> Fig. 12a

9. Page 22, par. 15, line 3: We -> we, mechanism -> mechanisms

10. Table A2: domain depth 100 -> 1000

---

## Author Response (AR1)

**Response to Referee report 1 on**

**Improving subduction interface implementation in dynamic numerical models**

Dan Sandiford          Louis Moresi

April 25, 2019

We thank the referee, Dr. Duretz, for the extensive and detailed comments on our manuscript. These have helped us to define the scope of the study more clearly, and also to the express some of the technical details more succinctly. In this reply, the comments of the referee (R1) appear in `typewriter font`, with our responses following directly beneath.

**General comments**

```
This study aims at providing a new approach to model long-term subduction processes,
particularly at the plate interface.  The authors investigate the widely-used weak
layer (WL) approach and identifies some of its limitations.  In this light, they propose
an alternative approach, termed embedded fault (EF), and show that this approach remedies
some limitations of the WL approach.  This work is interesting because many geodynamic
modelers studying subduction processes are facing these issues.  The outcome of this
study (i.e.  the EF approach) might help subduction modelers designing their models.
It will surely help informing the community about the caveats of subduction modeling.
That said, I was personally not convinced of using the proposed EF approach.  The
essence of this approach is to remap the geometry of the plate interface at each time
step of a simulation.  The idea is thus to overrule the geometries predicted by the
numerical simulation in order to facilitate interplate decoupling.  I would personally
not encourage code users to interfere within simulations by using ad hoc rules.  I
would rather expect an alternative solution that would not require interfering with
an ongoing numerical simulation.  This would necessitate an augmented modeling framework,
by accounting for thermo-mechanical feedbacks (e.g.  Thielmann and Kaus 2012) hydrological
(e.g.  Dymkova and Gerya 2013) or chemical processes) or an advanced rheological model
(e.g.  Bellas et al. 2018).  This is likely beyond the scope of the current study,
however this is a fundamental issue.  and these aspects are so far not really discussed.
Prior to publication, I would hence recommend the authors to enrich their discussion
- potentially around the above-mentioned points.  I found the discussion very short
and mostly restricted to the difference between WL/EF approaches.  I would also encourage
the authors to make sure that the figures are cited in increasing order.  Youll find
below a list of comments and suggestions.
```

The summary of R1 acknowledges that our study may help subduction modelers designing their models. We are encouraged by this reflection, as this was certainly one of our major intentions. The primary criticism of R1 here is somewhat philosophical - we would argue - and involves an alternative view on the best way forward for *improving subduction interface implementation in dynamic numerical models*. Referring to our proposed Embedded Fault strategy, R1 proposes

'an alternative solution that would not require interfering with an ongoing numerical simulation. This would necessitate an augmented modeling framework, by accounting for thermo-mechanical feed-backs, hydrological or chemical processes) or an advanced rheological model...'

R1 advocates the development of a more self-consistent approach to modelling the subduction interface, in which a range of feed-back mechanisms, including hydrological processes are implemented. In response, we simply suggest that there are valid reasons for pursuing both strategies. The suggestion of R1 likely involves significant increase in the complexity of models. Moreover, it is not clear that such as strategy is presently capable of allowing long-term, stable, buoyancy-drive subduction to operate. In cases where the aim of the subduction model is to understand, for instance, the long-term dynamics of the slab, such complexity may be undesirable. In other words, where a deliberately simplified representation of the subduction interface is desirable.

We also point out that the WL approach generally represents an ad hoc strategy, rather than an attempt to directly model the complex (multi) physics of the subduction interface zone. Given the ad hoc nature of WL implementations, it seems sensible to at least develop a more thorough understanding of the behavior and characteristics of that approach. This is precisely what our study intends to do.

The points we have briefly covered here have been integrated into the revised manuscript, particularly in Section 3.

**Specific comments**

`p.1 l. 8 – What is 'fully dynamic'?`

This term has been used in the literature to refer to a numerical subduction model driven only by internal buoyancy forces, rather than being additionally forced by velocity boundary conditions. In a number of places we have simplified to this phrase to 'dynamic models'. The term fully dynamic is clarified in Section 4.1.

`What are the requirements for a model to be 'fully dynamic'?`

See previous comment.

`p.2 l.5 – 'geodyanamics'`

Typo fixed.

`p.3 l. 15 – What is 'full thermal modelling'? How can a thermal model be 'full'?`

The word 'full' was unintended here, now corrected.

`p.5 l.9 – 'an solution'`

Typo fixed

`p.5 l.11 & p.5 l.25 – brackets around citation,`

Syntax fixed

`p.5 l.26 – a more important issue is when the interface locally thins out and becomes unresolved`

We agree that this is an important issue. We discuss this in Section 6.

`p.5 l.31 – ?`

Syntax fixed.

[Figure]

Figure 1: Vertical coordinate shows the vertical node number, for a 192 node mesh, with 192 being the surface node. Horizontal coordinate shows the element size. Blue line show the vertical element size in our highest resolution model.

 figure 1 caption:  The layer representing the subduction interface appears to have viscosity variations (top-left) while it is described as having a constant linear viscosity.

The rheology associated with the weak layer is merged with the background (plate/mantle) rheology at a significant distance (800 km) away from the trench on the subducting plate. The main reason for doing this is so that the weak layer does not extend all the way to the spreading ridge. This has been clarified in section 4.1. We think it is still appropriate to refer to the subduction interface as having a constant viscosity.

 Is figure 1 relevant at all, I dont see it called in the text?

Figure 1 has been significantly changed and is now referred to in the main text. We think that it is relevant to provide context for the type of simulations we are discussing, as well as to clarify some of the SZ terminology that is used throughout the paper.

 p.6 l.15 - psuedo-brittle

Typo fixed.

 p.6 l.24- what function you use to refine the mesh?

Given a initial mesh with N uniformly spaced nodes $(y_i)$, we shift the nodes to new locations $(y_i')$ using an normalised exponential function:

$$y_i' = (y_i - 1)e^{\lambda(\bar{Y}_i^2 - 1)} + 1$$

Where $\bar{Y}_i$ is the normalised position

$$\bar{Y}_i = \frac{-y_i - y_N}{y_N - y_0}$$

The constant $\lambda = 0.7$. Fig. 1 shows the element spacing based on this function.

`p.6 l.24-what is the horizontal resolution?`

All models have an aspect ratio of 5 which is reflected in the mesh resolution. So the highest-resolution models (192 elements in vertical dimension) have 960 elements in the horizontal dimension.

`p.6 l.30 - on all side`

Clarified.

`p.6 l.33-34 - not clear whether left and right hand side wall are treated similarly.`

The left and right hand side wall have the same boundary condition, this has been clarified.

`figure 2 caption:  Since you mention normal velocity to be zero at boundaries, you may also add that the tangential shear stress is zero.`

We have added this information, and fixed some typos in the Figure 2 caption.

`p.7 l.3 - 'Past studies...'  I would add references or delete this statement.`

Statement deleted.

`p.7 l.8 - 'Throughout this study...'  this sentence reads weird or incomplete.`

This section has been rewritten.

`p.8 l.1 - 'In this chapter' - is this part of a thesis or report?`

Fixed this mistake.

`p.8 l.4 - Fig.  8 is called right after Fig.  3`

Removed this figure reference.

`From Section 4.3 and Figure 3 caption, it is not entirely clear what the principle of EL is.  Do you mean that:  (1) you pre-define a channel geometry that will remains constant during model evolution (2) you remap, at each timestep, material types based on their relative position of the particles with regard to the channel?  If yes, please state it clearly.`

We mean (2). Section 4.3 and Figure 3 caption have been largely re-written to make this clearer.

`p.8 l.7 'In a number of previous studies...'  Do you mean variations in space or time?  No sure whether the referencing is sound.`

Primarily we mean spatially. We cannot say with certainly whether temporal variations are present in the studies we refer to, as this level of detail is not provided.

`p.  10 l.  20 'effected'`

Typo fixed.

`p.  10 l.  21 'these feature'`

Typo fixed.

`p.  10 from l.  25 on.  Better write $W_{\min}$ instead of $W_m in$.  Same for 'max' and 'init'`

Fixed.

`p.  10 l.30 please add a scale on this figure`

This figure has been updated with a scale.

```
p.  10 l.30 'physically inconsistent' do you mean geologically irrelevant?  Is this
so irrelevant by the way?
```

We mean, in a qualitative way, geologically irrelevant. The morphology shown in Figure 7. (to which p. 10 l.30 alludes) is dissimilar to current subduction zones on earth. One could make the argument that the extremely long forearc region ($\sim 500$ km) is also seen in some flat slabs settings (although arc volcanoes may not actually be present in those regions). However flat slabs usually have a subduction hinge with relatively normal curvature, whereas at greater distance from the trench the slab dip reduces and the curvature changes sign. In contrast, the model shown in Fig 7b, the length of the forearc is due to the extremely low curvature, but there is no slab dip reduction (flattening).

```
p 10.  l.34 - how would a free surface affect this behaviour?
```

While a free surface is likely to change some aspects of subduction dynamics, the variation in interface thickness - seen in the WL approach - would still be expected as the argument relating to variation in volume flux remains the same. We cannot guarantee that a model with a free surface would behave identically when the interface is not allowed to thicken (e.g. as in Fig 7 b).

```
figure 6 caption:  what are red and black dashed lines?  Which model is depicted
here?  The model presented in Fig.  7a or 7b?
```

Figure caption has been updated. The model depicted here is the same as in Fig.1a and Fig. 1b. (in the revised manuscript). This has been noted in the caption as well.

```
p.12 l.10 - reduce the amount the transient adjustment
```

We're unsure what the comment is.

```
p.12 . l.16 - 'One advantage
of the EF approach is that it offers improved precision in determining the thickness of
the subduction interface.' This is confusing, I thought the EL approach was aiming
at imposing this thickness. How can it help to determine a thickness when it already
imposes it?
```

Yes, by imposing a maximum and minimum thickness, we gain precision in the knowledge of its thickness, don't we?

```
p.12 l.17 - 'Such precision will be important for studying highly pressure- and temperature-sensiti
processes, such as metamorphism and melting near the slab top.'  Well, sure.  This
new parameterization will pre-define everything related to it.
```

See previous comment.

```
figure 7:  very difficult to appreciate the dimensions, a scale is missing.  What
is the grey line?  I think the notion of 'physically-consistent subduction morphology'
does not make sense.
```

Figure 7 has been substantially updated along with the caption. We have also incorporated R1's suggestion on 'physically-consistent subduction morphology', both here and in the main text.

```
figure 8: How is this MDD monitored? Do you measure the stress differences across
the plate interface?
```

The MDD is monitored by evaluating the vertical component of the velocity at a distance 5 km above the top of the slab (in a slab-normal sense). The vertical velocity clearly reveals the location of the onset of coupling.

```
figure 9:  do you use the same scale in x and y?  figure 9 caption:  'effected'
```

Figure 9 has been updated and typo fixed.

```
figure 10: at what time do these snapshots correspond? Do you use any particle
reseeding?
```

Snapshots are at 12.5 Myr, after the WL interface thickness has reached its quasi-equilibrium profile. This is now recorded in the figure caption. Particle reseeding is used to maintain regular particle numbers in the elements.

```
figure 10 - 11: Given the fact that the mesh is not distorted, adding element edges
would help the reader to realise how well the interface is resolved.
```

True, but as the mesh is quite high resolution we feel it would also detract from the clarity of the image. We have tried to convey this information (i.e. the interface resolution) where necessary.

```
p.15 l.9 - you mentioned you ran simulations with '72, 96, 128, 160 elements'. You
can also include 192 elements as I understand.
```

We did, and this has been corrected.

```
p. 16 l.5 - 'better' sounds very qualitative. You mean that low resolution models
using a WL fails at capturing plate decoupling.
```

Agreed. We have expressed this differently.

```
figure 13: I dont understand, why do repeated models at the resolution of 192 produce
any error? If the models were repeated you should obtain the exact same results,
dont you?
```

The only difference in the models is the randomness in initial particle locations, as well as the repopulation of those particles. We would note that the error, or drift, is still quite small. We are not aware of a similar experiment previously being published, so we cannot comment on whether this value is unexpectedly high.

```
p. 20 l.10 - a mistake here, a mean stress is not lithostatic. It can be split
into dynamic and lithostatic components.
```

This mistake has been corrected.

```
p. 22 - Why using a lithostatic pressure field in the viscosity expression. Are
obtained numerical solution of the actual pressure field not accurate enough to be
used in the rheology?
```

This is not indicative of poor results in the pressure calculation, but rather the assumption that the dynamic contribution of the pressure is expected to have to little effect on the viscous flow law compared to the lithostatic pressure contribution. In contrast, the dynamic pressure component is used in the determining the yield surface (Drucker-Prager criteria). The reviewer is correct that it would be more consistent to use the full pressure, however we doubt that the results of this study would be changed.

**References**

Thielmann, Marcel and Boris JP Kaus (2012). "Shear heating induced lithospheric-scale localization: Does it result in subduction?" In: *Earth and Planetary Science Letters* 359, pp. 1–13.

Dymkova, D and T Gerya (2013). "Porous fluid flow enables oceanic subduction initiation on Earth". In: *Geophysical Research Letters* 40.21, pp. 5671–5676.

Bellas, Ashley et al. (2018). "Dynamic weakening with grain-damage and implications for slab detachment". In: *Physics of the Earth and Planetary Interiors* 285, pp. 76–90.

**Response to Referee report 2 on**

**Improving subduction interface implementation in dynamic numerical models**

Dan Sandiford  Louis Moresi

April 26, 2019

We thank the referee for their critical reading of the manuscript, which raised a number of important issues and has helped us to clarify the central arguments in the study. In this reply, the comments of the referee (R2) appear in `typewriter font`, with our responses following directly beneath. Note that in our updated manuscript the figure order has changed slightly. Unless otherwise stated, figure numbers here refer to the first submitted manuscript version.

**General comments**

```
This paper explores different parameterizations of a weak layer that is used to model
one-sided subduction.  It particularly explores the controls on the width of this
weak layer and finds that, within the context of the models presented, there is a
preferential width.  In general I think this is a nice, albeit technical, contribution
to the literature describing fully dynamical modeling of subduction zones.  I have
a few somewhat broader comments and a fair bit of more detailed comments.
```

A continuously-entrained weak layer (WL) is a common strategy used in numerical models to provide decoupling between the slab and upper plate. In the models presented we show that significant thickness variations develop when a uniform thickness WL is initially prescribed. We argue that changes in the volumetric flux in the weak layer lead to these spatial and temporal variations in thickness. The comment that a preferential width is found 'in the context of the models presented', seems to imply that our results are model specific and not generalizable. In later parts of the review, R2 suggests that mesh resolution might be a contributing factor to the observed thickness variations and that such effects would be mitigated with increasing resolution.

We feel that these suggestions are made without any real justification, and do not critically engage with the argument we make. Our explanation for the evolution of the interface thickness invokes one main assumption, which is a first-order characterization of the kinematics of the WL, namely that uniform flow of material on the incoming subducting plate transitions to simple shear (Couette flow) within the decoupling region. We show that the interface evolves to an equilibrium thickness profile close to that predicted under this assumption. To help further clarify these processes, we introduced a simple boundary-driven example (e.g. fig. 5) which shows analogous thickness changes in a weak horizon due to a changing boundary condition and concomitant gradients in downstream volume flux. Hence, we have already gone to some effort to show that our analysis reveals an intrinsic behaviour of this kind of flow.

Of course, depending on other aspects of the model setup (e.g. the subduction interface rheology, use of adaptive meshing), width variations of the WL will effect the evolution of the simulation differently. However, the width variations themselves are unlikely to be specific to our models,

nor are they an artifact of low resolution. We feel that we may not have communicated these arguments as clearly as we could have, and have therefore re-written substantial parts of Section 5.1.

**Specific comments**

`Section 2 contains a fair bit of information that was already mentioned in the introduction or other superfluous. A bit of careful editing can shorten this section and make it clearer.`

Based on comments from all reviewers, we made minor adjustments to this section, with some simplifications.

`The citations in the text dont seem to follow any chronological or alphabetical order (see example in P4.l9-l11).`

This has been remedied.

`The weak fault seems rather wide (10 km as mentioned on page 7). If you have a 2 km grid resolution then that it is probably a good idea to smear it out over such a distance but what are your thoughts about FEM grid refinement around this fault to be able to bring it to more realistic thicknesses?`

We chose a WL initial thickness that was representative of recent long-term numerical subduction models, e.g. (e.g. Čížková, Hunen, and Berg 2007; Garel et al. 2014; Agrusta, Goes, and Hunen 2017). Again, as our paper is methodological, we are content with using a model setup that is appropriate for the problem. Of course, in the highest-resolution models we showed (e.g. element width of $\sim 2.5$. km) we could resolve a thinner interface. But we doubt that doing so would alter any of the conclusions of the study.

Based on our analysis of the resolution convergence, we would suggest that the ratio of prescribed WL thickness to element size should be greater than three. This is not to say that a model with less resolution won't produce physically consistent subduction, however it may not show satisfactory resolution convergence. Testing resolution convergence is obviously the best way to ascertain this for a specific setup. We argue that in the WL approach, there is a component of 'wasted' resolution that is required only to properly resolve the transient phase of interface adjustment. This is why, we argue, the error accumulation tends to decrease significantly at around the time the interface thickness reaches quasi-equilibrium (see Figure 13.).

`I wonder if the improvements you find when choosing the initial and maximum width of the weak zone to be 20 km is just a matter of the numerical resolution (mesh and particles) that you use. Maybe you need at least 10 elements in this zone for things to stabilize. You mention that a lot of small wavelength variations disappear compared to using 10 km. This might be a symptom of discretization issues...`

The improvements we mention (using the EF approach) are specifically improvements relative to a standard WL setup. We assessed both implementations at several resolutions. We showed that the EF approach has better resolution convergence. This means that as we increase the resolution, the error (relative to a high-resolution reference model) is always smaller in the EF approach than in an equivalent WL approach. We expect that further adjustments to the EF approach (or similar strategies) could improve this further. R2 is correct to point out that the EF method always starts out with a higher effective resolution in the decoupling region - hence the comparison between the EF and WL approaches does have a limitation.

The EF approach has two separate features, one is the imposition of a variable initial thickness, the other involves controlling the maximum and minimum thickness. Both of these are targeted

at different behaviours we observed in the WL model. The first tries to limit the amount of transient adjustment of the interface thickness, the second helps to dampen the short-wavelength instabilities at the boundary of the subduction interface and the upper plate. We could, of course, investigate these aspects individually. However, our main goal here was to design and communicate an approach that achieves better outcomes (e.g. better resolution convergence) for a given computational expense. We think that the EF approach achieves this goal.

```
It would be very helpful if you could repeat your exercise with higher resolution.
I suspect your maximum width will go down with increasing resolution.  In 5.3 you
do a divergence test but I am not sure that this is very meaningful given that you
show that coarser resolution meshes lead to worse results.  It is far more exciting
to see what happens when you start converging.
```

As we discuss in our reply to the preceding comment, there are some limitations in the convergence (divergence?) test, and therefore some ambiguity in how these tests are interpreted. We hope that we have already answered these criticisms satisfactorily. The second point made here by R2 is that the pattern of width variations in the WL approach might be resolution dependent, and that such processes would be mitigated with increasing resolution.

Our reply here essentially follows from our answer to the comment in paragraph 1. The width variation are not likely to be a dependent on resolution or 'context' (see paragraph 1), because the width variations are an intrinsic outcome of the fluid dynamics (see reply to paragraph 1).

The referee has provided no physical argument to back up the expectation that these processes should be resolution dependent. Hence, we are skeptical that it would be helpful to reproduce our results with higher-resolution models.

We accept that we may not have communicated these arguments as clearly as we could have, and have therefore re-written substantial parts of Section 5.1.

```
Figure 11.  I am not certain it is useful to show under-resolved results.
```

In this case we feel it is useful, because the style in which the models degenerate actually reveals something about the way that the interface thickness is dependent on the kinematics of the flow. In particular, when the interface is under-resolved and does not adequately decouple the plates, it is the induced partial-coupling that drives further thinning of the interface. This feedback cycle is a limitation of the WL method, and may be very relevant when additional processes are included - such as subduction of buoyant crust / continents.

```
P1 footnote 1.  why footnote?
```

We have removed the footnote.

```
might be good to add a few original references as to why the slab-mantle wedge coupling
appears to start at 80 km (that seems to be the case in most SZs; please just clarify
why we think this is the case)
```

We have cited the paper of Wada and Wang 2009. In the scope of this study we feel this sufficient.

```
P2.l5.  typo
```

Typo fixed.

```
P2.l25.  typo
```

Citation fixed.

```
P2.l25-27.  Full sentence here.  You can probably rephrase this a bit better as in
that sediments may be important amidst various other issues controlling plate velocities.
```

We have reorganised this paragraph.

```
P2.l33.  10s -> tens
```

Changed.

P3.l12-14. I am confused about this full sentence here. The seismogenic zone is generally not characterized by ending in the mantle (maybe between slab crust and mantle, but not slab mantle and mantle).

We have simplified this section.

P3.l14. Average stresses: where?

The value we quote (from Lamb 2006) is based on an average stress taken along the entire subduction interface.

P3. line starting at l22. This seems to take a single point of an antigorite-out boundary as some extreme. Its a bit more variable/complex than that. You can probably delete this sentence.

We have reconfigured this paragraph substantially based on the comment.

P4. Figure 1. last line 'representative' of what?

Figure 1 has been changed and captions and labels redone.

P4.l14 missing article

Changed.

P4.l5 missing plural

This paragraph has been condensed.

P4.l9 typo

Changed.

P5.l11 typos

Fixed.

P5.l25. What is 'over-resolving?

We mean providing resolution larger than is necessary to resolve an interface of a given thickness. This would ensure that the interface remains adequately resolved when the interface thins (as occurs in the deeper part of interface under WL approach - e.g. Fig. 4).

P5.l30 just benchmarking

Changed

P5.l31. '1 km, 20K' Not sure what this means. The benchmark study referenced here showed that finite element models agreed on temperature predictions along the slab surface to within about 1K for 1 km resolution. There were some finite difference models that had larger differences.

We have shortened the paragraph and excised this sentence.

And here and below. It is just 20K not 20°K.

Fixed.

P7 three sentences starting at l20. Dont you say essentially the same thing here? You might be able to condense this into a single sentence.

We have simplified this paragraph.

P8.l1 I hope a cosine taper is easy to implement in many subduction models. Is it not?

Past studies have used different approaches to effectively inhibit the subduction interface from extending to arbitrary depths. In the context of the Lagrangian particle method, we could simply (arbitrarily) replace the weak layer material with background material at a given depth. We choose a rheological transition, based on a cosine taper, to allow a somewhat graduated change.

`P9.l12ff. Maybe use subscript 'conv?`

We have simplified the mathematical symbology in this section.

`P9.bottom. MDD reference is repetitive`

Changed.

`P10.l6 You are citing Figure 8 way out of order`

Yes. We have omitted this reference.

`P10.l21 typo`

Fixed.

**Response to Referee report 3 on**

**Improving subduction interface implementation in dynamic numerical models**

Dan Sandiford          Louis Moresi

April 25, 2019

We thank the referre, Dr. Čížková, for providing a number of helpful comments on our manuscript. In this reply, the comments of the referee (R3) appear in `typewriter font`, with our responses following directly beneath. Note that in our updated manuscript the figure order has changed slightly. Unless otherwise stated, figure numbers here refer to the first submitted manuscript version.

**General comments**

```
Manuscript presents a detailed analysis of the properties of entrained weak layer
that is often used in numerical models of subduction to decouple the plates.  The
authors demonstrate that this weak layer is changing its width both spatially and
with time.  After entering the subduction channel at the trench it increases width
and then thins again as the subducting plate is coupled with the mantle flow below
the overriding plate.  This thinning, that may temporarily result in a decoupling
weak layer actually yet thinner than the prescribed initial thickness, may result
in locking of the subduction in case of lower resolution and certainly affects the
subduction evolution.  The authors explain this phenomenon using an analogy with a
boundary-driven Stokes flow in a two layer material with boundary condition changing
from free-slip to no-slip and back.
```

```
They then introduce an improvement to the standard weak layer approach (called here
embedded fault) where the thickness of the weak layer is controlled and modified during
the subduction evolution.  This approach prevents transient thinning of the lower
part of the decoupling layer and potential coupling of the subducting and overriding
plates incase of lower resolution.
```

```
I find this paper very interesting.  The weak layer approach is often used in subduction
modelling while the numerical aspects of the implementation of this decoupling are
seldom discussed or even mentioned.  I therefore very much appreciate this systematic
evaluation of the problem and suggested solution using ad hoc control of the layer
thickness.  The crutial point of the paper is made in fig.  12 where the authors illustrate
that in their embedded fault approach low resolution has much smaller effect than
in standard weak layer approach.  This paper thus provides the reader with a recipe
how to tackle the problem of decoupling the plates with an entrained weak layer  one
may either use the suggested embedded fault approach, or use resolution high enough
to resolve the thinned interface in the transient stage at the beginning of subduction.
Alternatively, as just briefly mentioned in the discussion, this problem may be suppressed
by nonlinear rheology, but that of course brings other complexities into the play
```

The manuscript is nicely and clearly written and the topic fits the scope of Solid
Earth,therefore I recommend it for publication.  I only have couple of suggestions
for mostly minor corrections.

As the general comments above mainly summarise our paper, and are largely supportive of the
scope and conclusions, we thank the reviewer and simply address the specific points listed below.

**Specific comments**

You may perhaps explicitly mention that the weak layer has constant viscosity.  The
reader has to dig up this information by combining the sentence stating that except
of subduction interface the rest of the domain is deforming according to composite
rheology (at the end of paragraph 20 page 22 in Appendix) and the information in Table
A2 (unless I overlooked this information somewhere else).

In the updated manuscript we have made additional reference to the subduction interface viscosity
in Section 3 paragraph 2, as well as Section 4.2.

I dont see the logic in the order of figures  they are sometimes ordered and referenced
rather randomly.  Figure 1 is not referenced in the text.

Figure 1 has been substantially changed and is now referenced in the text. Other figure references
that were out of order have been rectified, while the figure order has been changed slightly.

Page 2, par.  25, line 4:  remove of

Typo fixed.

Page 5, par.  5, line 5:  an solution -> a solution

Typo fixed.

Page 5, par.  10, line 3:  though -> thought

Typo fixed.

Page 6, par.  5, line 3:  a initial -> an initial

Typo fixed.

Page 6, par.  20, line 5:  a element -> an element

Typo fixed.

Caption Figure 11:  Fig.  a -> Fig.  12a

Fixed.

Page 22, par.  15, line 3:  We -> we, mechanism -> mechanisms

Typo fixed.

Table A2:  domain depth 100 -> 1000

Fixed.

**Improving subduction interface implementation in dynamic numerical models**

Dan Sandiford[1,2] and Louis Moresi[1]

[1]School of Earth Sciences, University of Melbourne, VIC, 3010, Australia
[2]Institute of Marine and Antarctic Studies, University of Tasmania, TAS, 7004, Australia

**Correspondence:** Dan Sandiford (dan.sandiford@utas.edu.au)

**Abstract.**  Numerical subduction models often implement an entrained weak layer (WL) to facilitate decoupling  of the slab and upper plate. This approach is attractive in its simplicity, and can provide stable, asymmetric subduction systems that persist for many tens of Myr. In this study we undertake a methodological analysis of the WL approach, and use these insights to guide improvements to the implementation. The issue that primarily motivates the study is the emergence of significant spatial and temporal thickness variations within the WL. We show that these variations are mainly the response to volumetric flux gradients, caused by the change in boundary conditions as the WL material enters and exits the zone of decoupling. The time taken to reach a quasi-equilibrium thickness profile will depend on the total plate convergence, and is around 7 Myr for the models presented here. During the transient stage, width variations along the WL can exceed 4×, which may impact the effective strength of the interface,  through physical effects if the rheology is linear,  or simply if the interface becomes inadequately numerically resolved. The transient stage also induces strong sensitivity to model resolution. By prescribing a variable thickness  WL at the outset of the model, and by controlling the limits of the layer thickness during the model evolution, we find improved stability and resolution convergence of the models.

**1 Introduction**

~~that are simple enough to allow a tractable study of these phenomena, yet rich enough to be quantitatively constrained by the observations remains challenging (Buiter and Ellis, 2016). The subduction interface is a critical and sensitive component in such models. It must simultaneously provide strong strain localisation, low stress, lateral translation, and long term stability (Gurnis and Hager, 1988; Arcay, 2012).~~

5  The process of stable asymmetric subduction requires that the down-going plate is substantially decoupled [1] from the overriding plate along the subduction interface (Gerya et al., 2008). ~~In the majority of subduction zones, decoupling transitions rapidly to complete slab-mantle coupling at a depth of around 80 km (Wada and Wang, 2009). This means that the depth extent of decoupling is significantly greater than the limit of the locked megathrust zone (< 50 km). The subduction interface zone is characterised by rheological and petrological complexity, low strength (relative to the slab), as well as the abundance of water~~

[revised manuscript text omitted]

is too strong, subduction will be unrealistically slow, or may entrain the upper plate in a mode of catastrophic buckling and failure Despite the fact that we can draw analogies to relevant physical processes, our view is that the WL approach constitutes an ad hoc solution to the sub-grid phenomena of plate boundary shear localisation.

The behavior of the WL is discussed is some detail in Arcay (2012, 2017). These studies highlight the tendency for the subduction interface to develop spontaneous thickness variation as the models evolve. Typically the interface widens near the trench, building a prism-like complex, and thins at depths beyond the brittle-ductile transition ( 50 km). This pattern was also noted in the boundary element models of (Gerardi and Ribe, 2018), who attributed a downdip thickness variation to lubrication layer dynamics. The tendency for thinning will impact the level of mesh resolution required in a model (Arcay, 2017). Additionally, WL formulations are likely to evolve maximum subduction interface thickness around twice the imposed thickness, potentially impacting effective stress along the interface, as well as the accuracy of predicted thermal structure. Understanding and controlling these spontaneous width variations will be important in terms of using dynamic models to explore sensitive subduction-zone processes, such as metamorphism and melting near the slab top. Currently, these type of questions are mainly addressed using thermo-kinematic models. Bench-marking of community codes indicates that precision to about 1 km, & 20°K can be expected in well resolved thermo-kinematic models (van Keken et al., 2008). This provides a useful, if ambitious target for dynamic numerical subduction models.

[revised manuscript text omitted]

5 ~~variations. The first can be understood as an outcome of the evolution towards uniform volumetric flux along the interface. This phenomenon is likely to be present in any implementation of the WL approach. The second represents shorter-wavelength instabilities at the boundary between the subduction interface and the upper plate. Both of these characteristic behaviours can be seen in Fig. 4, where we plot the interface thickness as a function of the downdip distance during 20 million years of model evolution. During the first few million years of the model evolution, changes in the interface thickness have both spatial and~~

10 ~~temporal dependency. The interface thickens at distances of up to 150 km downdip from the trench, whereas beyond this it thins. The point of zero thickness change (10 km) migrates downdip with time.The interface reaches a minimum thickness of around 5 km at ~ 2.0 Myr, while it reaches 20 km, close to its maximum thickness, at around 7 Myr. The thickened part of the interface evolves to a near-constant configuration, both in thickness (~ 20 km) and downdip extent (~ 200 km) . In contrast the thinning of the deeper interface is transient and by ~ 15 Myr the deeper part of the interface has reestablished a thickness close~~

[revised manuscript text omitted]

Ulmer, P., Trommsdorff, V., et al.: Serpentine stability to mantle depths and subduction-related magmatism, Science, pp. 858–858, 1995.

30  van Keken, P. E., Currie, C., King, S. D., Behn, M. D., Cagnioncle, A., He, J., Katz, R. F., Lin, S.-C., Parmentier, E. M., Spiegelman, M., et al.: A community benchmark for subduction zone modeling, Physics of the Earth and Planetary Interiors, 171, 187–197, 2008.

Vannucchi, P., Remitti, F., and Bettelli, G.: Geological record of fluid flow and seismogenesis along an erosive subducting plate boundary, Nature, 451, 699, 2008.

Vannucchi, P., Sage, F., Phipps Morgan, J., Remitti, F., and Collot, J.-Y.: Toward a dynamic concept of the subduction channel at erosive
35  convergent margins with implications for interplate material transfer, Geochemistry, Geophysics, Geosystems, 13, 2012.

Vrolijk, P.: On the mechanical role of smectite in subduction zones, Geology, 18, 703–707, 1990.

Wada, I. and Wang, K.: Common depth of slab-mantle decoupling: Reconciling diversity and uniformity of subduction zones, Geochemistry, Geophysics, Geosystems, 10, 2009.

Wang, K.: Finding fault in fault zones, Science, 329, 152–153, 2010.

Watts, A. and Zhong, S.: Observations of flexure and the rheology of oceanic lithosphere, Geophysical Journal International, 142, 855–875, 2000.

Zhong, S. and Gurnis, M.: Viscous flow model of a subduction zone with a faulted lithosphere: Long and short wavelength topography gravity and geoid, Geophysical Research Letters, 19, 1891–1894, https://doi.org/10.1029/92gl02142, http://dx.doi.org/10.1029/92gl02142, 1992.

Zhong, S. and Gurnis, M.: Mantle Convection with Plates and Mobile Faulted Plate Margins, Science, 267, 838–843, https://doi.org/10.1126/science.267.5199.838, http://dx.doi.org/10.1126/science.267.5199.838, 1995.